# NRK1 controls nicotinamide mononucleotide and nicotinamide riboside metabolism in mammalian cells

Joanna Ratajczak[1,2], Magali Joffraud[1], Samuel A.J. Trammell[3], Rosa Ras[4,5], Núria Canela[4,5], Marie Boutant[1], Sameer S. Kulkarni[1], Marcelo Rodrigues[3,6], Philip Redpath[6], Marie E. Migaud[3,6], Johan Auwerx[7], Oscar Yanes[5,8,9], Charles Brenner[3] & Carles Cantó[1,2]

$NAD^+$ is a vital redox cofactor and a substrate required for activity of various enzyme families, including sirtuins and poly(ADP-ribose) polymerases. Supplementation with $NAD^+$ precursors, such as nicotinamide mononucleotide (NMN) or nicotinamide riboside (NR), protects against metabolic disease, neurodegenerative disorders and age-related physiological decline in mammals. Here we show that nicotinamide riboside kinase 1 (NRK1) is necessary and rate-limiting for the use of exogenous NR and NMN for $NAD^+$ synthesis. Using genetic gain- and loss-of-function models, we further demonstrate that the role of NRK1 in driving $NAD^+$ synthesis from other $NAD^+$ precursors, such as nicotinamide or nicotinic acid, is dispensable. Using stable isotope-labelled compounds, we confirm NMN is metabolized extracellularly to NR that is then taken up by the cell and converted into $NAD^+$. Our results indicate that mammalian cells require conversion of extracellular NMN to NR for cellular uptake and $NAD^+$ synthesis, explaining the overlapping metabolic effects observed with the two compounds.

[1] Nestlé Institute of Health Sciences (NIHS), Lausanne CH-1015, Switzerland. [2] School of Life Sciences, Ecole Polytechnique Fédérale de Lausanne (EPFL), Lausanne CH-1015, Switzerland. [3] Department of Biochemistry, Carver College of Medicine, University of Iowa, Iowa City, Iowa 52242, USA. [4] Group of Research on Omic Methodologies (GROM), Universitat Rovira i Virgili, Reus 43204, Spain. [5] Centre for Omic Sciences, Universitat Rovira i Virgili, Reus 43204, Spain. [6] School of Pharmacy, Queen's University Belfast, Belfast BT7 1NN, UK. [7] Laboratory of Integrative and Systems Physiology, Ecole Polytechnique Fédérale de Lausanne (EPFL), Lausanne CH-1015, Switzerland. [8] Biomedical Research Centre in Diabetes and Associated Metabolic Disorders (CIBERDEM), Madrid 28029, Spain. [9] Department of Electronic Engineering, Universitat Rovira i Virgili, Tarragona 43007, Spain. Correspondence and requests for materials should be addressed to C.B. (email: charles-brenner@uiowa.edu) or to C.C. (email: carlos.cantoalvarez@rd.nestle.com).

Nicotinamide adenine dinucleotide ($NAD^+$) is an essential cofactor for multiple cellular redox processes linked to fuel utilization and energy metabolism, including the mitochondrial oxidative phosphorylation system[1,2]. $NAD^+$ is also a substrate for multiple enzymes, including sirtuins and poly(ADP-ribose) polymerases (PARPs), which hydrolyse the glycosidic bond between the nicotinamide (NAM) and ADP-ribosyl moieties of $NAD^+$ (ref. 3). In the case of sirtuins, this reaction is coupled to protein deacylation, which yields the deacylated protein plus NAM and a mixture of 2'- and 3'-O-acyl-ADPribose as products. On the basis of the high $K_m$ for $NAD^+$ of some sirtuins, these enzymes have been postulated to act as $NAD^+$ sensors, translating metabolic cues into enzymatic and transcriptional adaptations[3]. Importantly, sirtuin activation has been associated with an improvement in metabolic efficiency and healthspan in mammals[4,5].

The relevance of $NAD^+$ precursors in health is further illustrated by the historical use of nicotinic acid (NA) and NAM, commonly known as niacin or vitamin $B_3$, in prevention and treatments for pellagra and dyslipidemia[6]. A limitation in the pharmacological use of niacin against dyslipidemia is the occurrence of unpleasant side effects in the form of flushing, mostly due to activation of the GPR109A receptor by NA[7]. NAM is much less efficient than NA as a lipid lowering agent, likely because NAM exerts product inhibition on sirtuin activity[1]. The metabolism of these conventional niacin compounds to $NAD^+$ is also different, as NA is converted via the three-step Preiss–Handler pathway, whereas NAM is metabolized into nicotinamide mononucleotide (NMN) via nicotinamide phosphoribosyltransferase (NAMPT) and then by NMN adenylyltransferases (NMNAT1-3) into $NAD^+$ (ref. 8).

As a phosphorylated compound, NMN is not a vitamin precursor of $NAD^+$ but rather a biosynthetic intermediate[2]. However, administration of NMN to mice produces multiple beneficial effects. It improves insulin sensitivity in high-fat diet-induced diabetes[9], protects the heart from ischaemia-reperfusion injury[10] and restores mitochondrial function in aged muscles[11,12]. Though the phosphoribosyl pyrophosphate substrate of NAMPT has not been detected extracellularly[13], it has been speculated that NMN might be a circulating $NAD^+$ precursor, produced by a circulating extracellular NAMPT activity[14]. However, there are discordant reports on circulating NMN concentrations, which have been reported to be $< 50$ nM by LC-tandem mass spectrometry[13] to $> 80 \mu M$ (ref. 14). The mechanisms by which extracellular NMN is converted to cellular $NAD^+$ also remain elusive. On the one hand, it was claimed that NMN is transported intact to hepatocytes[9]. On the other hand, it has been proposed that extracellular dephosphorylation of NMN to nicotinamide riboside (NR) is required to elevate cellular $NAD^+$ metabolism[1,15–17].

NR is a recently described $NAD^+$ precursor that might overcome the problems linked to the administration of higher doses of NA and NAM[18]. NR boosts $NAD^+$ synthesis and sirtuin activation without affecting GPR109A[19]. NR supplementation increases lifespan in yeast, Caenorhabditis elegans and mice[20–22] and improves high-fat diet-induced metabolic complications in mice[19,23]. NR has recently been shown to be the favoured orally available hepatic precursor to $NAD^+$ in mice and to safely boost human $NAD^+$ metabolism in single oral doses[24].

NR naturally occurs in milk[18,25] and its conversion to $NAD^+$ is initiated by phosphorylation of NR to NMN by NR kinases (NRKs)[18]. NRKs, encoded by the Nmrk genes, are highly conserved enzymes in all eukaryotes[18]. In mammals there are two NRK enzymes, NRK1 and NRK2, but little is known about their physiological roles. In this study, we explore how modulation of NRK activity influences the action of $NAD^+$ precursors. For this

purpose, we created NRK gain- and loss-of function cellular models as well as an Nmrk1-deficient mouse model (NRK1KO). Our results indicate that NRKs are rate-limiting for NR metabolism in mammalian cells, and also a required enzymatic activity for conversion of exogenously administered NMN to $NAD^+$.

## Results

**NRKs are rate-limiting for NR/NMN-driven $NAD^+$ synthesis.** We took advantage of the low endogenous NRK1 and NRK2 levels in NIH/3T3 cells (Fig. 1a,b) to test how increased NRK expression would affect NR-driven $NAD^+$ production in mammalian cells. For this purpose, we generated constructs to overexpress murine NRK1 or NRK2 enzymes, alongside predicted catalytically inactive NRK1-D36A, NRK1-E98A, NRK2-D35A and NRK2-D100A mutants[26]. NRK1 and NRK2 constructs were greatly overexpressed in NIH/3T3 cells with respect to endogenous mRNAs (Fig. 1a). However, NRK1-E98A and NRK2-D100A mutant proteins did not accumulate appreciably, indicating that these mutant proteins might be unstable (Fig. 1b). For this reason, we only used the NRK1-D36A and NRK2-D35A mutants for further experiments. As predicted by the low basal NRK1 and NRK2 expression in NIH/3T3 cells, empty vector-transfected cells did not display marked differences in $NAD^+$ levels in response to NR treatment (Fig. 1c). Also, overexpression of active NRK1 or NRK2 had no effect on basal intracellular $NAD^+$ content but was sufficient to convert NIH/3T3 from an NR-nonresponsive cell line to an NR-responsive cell line with increases in $NAD^+$ of $\sim 4$-fold (Fig. 1c). The assay further indicated that NRK1-D36A and NRK2-D35A mutants possess no kinase activity, as NR failed to increase $NAD^+$ levels when these mutant forms were expressed (Fig. 1c). Thus, these data establish that increasing NRK activity by transient overexpression is sufficient to boost NR-driven $NAD^+$ synthesis, suggesting a rate-limiting role for NRK1 and/or NRK2 in NR utilization.

To modulate NRK isozyme expression stably and at a lower level of overexpression, we generated stable NIH/3T3 cell lines that integrated single copies of the Nmrk1 (F3T3-NRK1) or Nmrk2 (F3T3-NRK2) genes, encoding for Flag-tagged NRK1 and NRK2 proteins, repectively. This strategy led to detectable increases in NRK1 protein levels (Fig. 2a) that were achieved despite a 500-fold reduction in mRNA levels with respect to transient overexpression systems (Figs 1a and 2a). By western blot, NRK1 was effectively overexpressed but NRK2 was difficult to detect, despite robust increases at the mRNA level (Fig. 2a). Given the pronounced tissue-specific expression of the NRK2 protein in mouse (Supplementary Fig. 1a,b), we considered the possibility that NRK2 requires muscle-specific factors for stability and therefore focused our efforts on NRK1. NRK1 protein levels in F3T3-NRK1 were comparable to the endogenous levels of NRK1 in tissues with high NRK1 expression, such as liver and kidney (Supplementary Fig. 1c). F3T3-NRK1 cells showed very significant dose-dependent increases in $NAD^+$ accumulation when NR was added to standard, NAM-containing DMEM at concentrations as low as $100 \mu M$, reaching maximal levels at $\sim 1$ mM concentrations (Fig. 2b). Of note, despite the modest expression of NRK2 in F3T3-NRK2 cells, this construct drove significant increases in NR-induced $NAD^+$ synthesis (Fig. 2b).

We also generated a variant of the F3T3-NRK1 cell line in which the catalytically inactive form of NRK1 was expressed (F3T3-D36A) (Fig. 2a). As expected from transient overexpression experiments, NR-induced $NAD^+$ synthesis was fully blunted in F3T3-D36A cells compared with the large increases observed in F3T3-NRK1 cells (Fig. 2d). We next

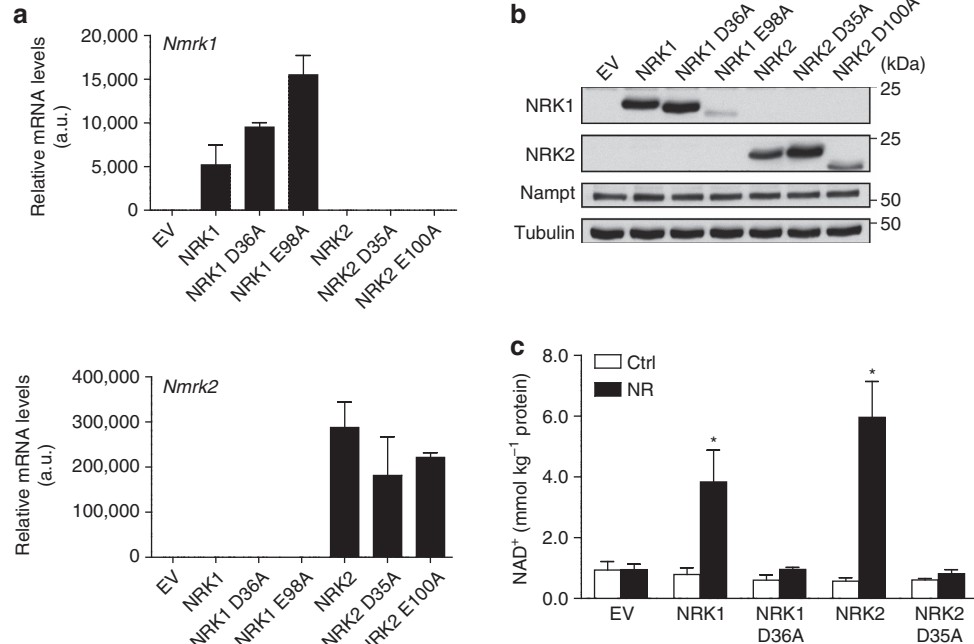

**Figure 1 | NRKs are rate-limiting for NR-driven NAD$^+$ synthesis.** (**a,b**) NIH/3T3 cells were transfected with 4 µg of Flag-tagged WT and catalytically inactive version of NRK1 or NRK2 overexpressing vector. *Nmrk1* and *Nmrk2* mRNA (**a**) and NRK1/NRK2 protein (**b**) expression. (**c**) NAD$^+$ measurement in NR-treated NIH/3T3 cells overexpressing NRK1 or NRK2. Results shown are mean ± s.e.m.; EV, empty vector; *$P < 0.05$ versus ctrl by two-sided unpaired *T*-test ($n = 3$).

evaluated the role of NRK1 in the utilization of other NAD$^+$ precursors, such as NMN and NAM. If NMN has the capacity to enter cells by diffusion or a specialized transporter, it would only require one enzymatic step (NMNAT1-3) to generate NAD$^+$ and, therefore, its ability to increase NAD$^+$ synthesis should be independent of NRK1 (Fig. 2c). On the other hand, if NMN is dephosphorylated prior to cell entry, then NMN should show dependence on NRK expression to form intracellular NAD$^+$. Consistent with a requirement of NMN to be dephosphorylated to NR extracellularly, NMN could only increase NAD$^+$ synthesis in F3T3-NRK1 cells, and less potently than NR (Fig. 2e). In contrast, NAM enhanced NAD$^+$ synthesis irrespective of NRK1 overexpression (Fig. 2f), illustrating that NRK1 is dispensable for NAM action. Considering that neither NAMPT nor NMNAT isozyme expression were reduced in our NRK1 transgenic F3T3 models (Supplementary Fig. 1d), these data indicate that NRK1 expression mediates NMN-driven NAD$^+$ synthesis in 3T3 fibroblasts. Importantly, NRK1 overexpression did not affect the expression of NADases, such as CD38 and its homolog CD157, or the CD73 nucleotidase, all of which have been reported to influence NMN conversion to NR[15,27] (Supplementary Fig. 1d).

**Extracellular conversion of NMN to NR by hepatic cells.** We next aimed to study NR and NMN metabolism in cell models of the greatest physiological relevance. For this reason, we first analysed NRK1 expression in mouse tissues. The *Nmrk1* gene is ubiquitously expressed at the mRNA level (Supplementary Fig. 1a) and the liver, an organ mediating many of the metabolic effects of NR[19,23,28], displayed high NRK1 protein levels (Fig. 3a). When we injected mice with 500 mg kg$^{-1}$ of NR intraperitoneally, NR elevated NAD$^+$ levels in liver, muscle, brown adipose tissue and brain within 60 min (Fig. 3b). Consistent with high level expression of NRK1 in liver, the fold-change of increased NAD$^+$ was greater in liver than in

other tissues (Supplementary Fig. 2a). This result suggests the possibility that tissues might have selective preferences for different NAD$^+$ precursors to sustain NAD$^+$ levels[2]. To test this hypothesis, we examined the sensitivity of different cell types to NAMPT inhibition. Thus, we treated primary hepatocytes, primary brown adipocytes and the muscle cell line C2C12—representing liver, brown adipose tissue (BAT) and muscle, respectively—with FK866, a NAMPT inhibitor. While, after 30 h of treatment, FK866 treatment decreased intracellular levels by more than 80% in differentiated brown adipocytes and C2C12 myotubes, primary hepatocytes still retained ∼50% of their endogenous NAD$^+$ levels (Supplementary Fig. 2b). The higher sensitivity of brown adipocytes and myotubes to FK866 could already be appreciated after only 6 h of treatment. Altogether, these results suggest that different tissues have characteristic rates of flux through the NAM salvage pathway and the NRK pathway, which bypasses NAMPT. Importantly, FK866 did not alter the response of these cell lines to NR or NMN, illustrating how these compounds are metabolized into NAD$^+$ in a NAMPT-independent manner (Supplementary Fig. 2c).

Based on the ability of liver to convert supplemented NR to NAD$^+$, we focused on liver-derived cell lines for further mechanistic analysis. NR treatment led to significant increases in NAD$^+$ content in HepG2, Hepa1.6 and AML12 cells (Fig. 3c,d and Supplementary Fig. 2d). Western blot analyses in Hepa1.6 and AML12 confirmed a correlation between NRK1 levels and the ability of NR to enhance NAD$^+$ synthesis (Supplementary Fig. 2e). Consistent with a rate-limiting role of NRK1, *Nmrk1* overexpression in AML12 or HepG2 cells was sufficient to boost the effect of NR on NAD$^+$ synthesis (Fig. 3c,d).

To more closely compare the effect of NR and NMN on the intracellular NAD$^+$ metabolome in cells that endogenously express NRK1, we utilized HepG2 cells to perform stable isotope tracer analyses. For this purpose, we labelled the carboxamide oxygen of NR and NMN with $^{18}$O and treated HepG2 cells

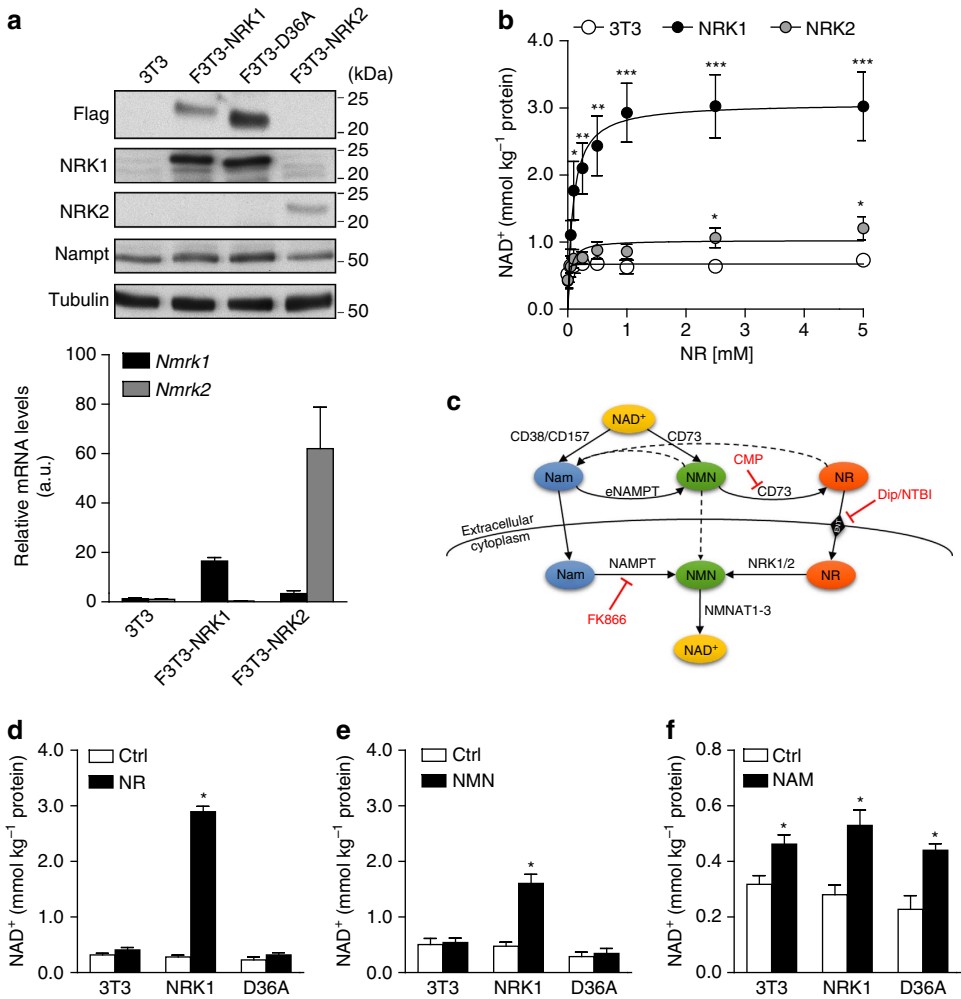

**Figure 2 | Slight overexpression of NRKs potentiates NR- and NMN-driven NAD$^+$ synthesis.** (**a**) NRK1 and NRK2 protein expression and mRNA levels in NIH/3T3 and stable cell lines with one additional copy of either of NRK (F3T3 cells). (**b**) NAD$^+$ measurement in NR dose-response treatment of F3T3 cells. (**c**) Schematic of NAD$^+$ precursors metabolism (**d–f**) NAD$^+$ measurement in F3T3 cells treated for 6 h with 0.5 mM NR (**d**), 0.5 mM NMN (**e**) and 5.0 mM NAM (**f**). Results shown are mean ± s.e.m., *$P < 0.05$, **$P < 0.01$, ***$P < 0.001$ versus ctrl by two-sided unpaired $T$-test ($n = 3$).

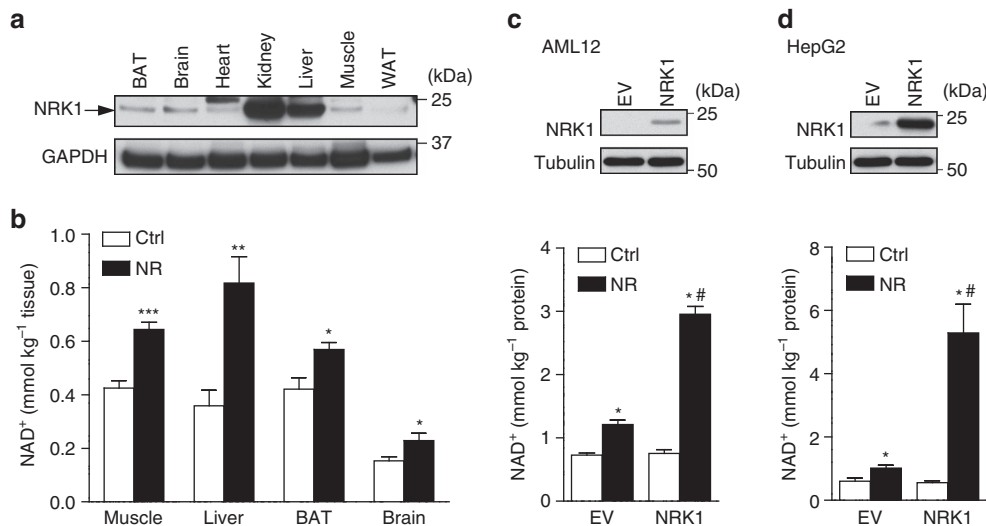

**Figure 3 | NRK1 and NR metabolism in hepatic cells.** (**a**) Protein levels of NRK1 in mouse tissues. (**b**) 7–9 weeks old WT male mice were IP injected with 500 mg kg$^{-1}$ of NR or vehicle and tissues were collected after 1 h. NAD$^+$ measurement in selected tissues, $n = 5$ per group. (**c,d**) AML12 (**c**) and HepG2 (**d**) cells were transfected with NRK1-overexpressing vector. NRK1 protein levels and NAD$^+$ measurement after 6 h 0.5 mM NR treatment. Results shown are mean ± s.e.m., *$P < 0.05$, **$P < 0.01$, ***$P < 0.001$ versus ctrl; #$P < 0.05$ versus EV by two-sided unpaired $T$-test or (in **c** and **d**) one-way ANOVA followed by Bonferroni's *post-hoc* test ($n = 3$).

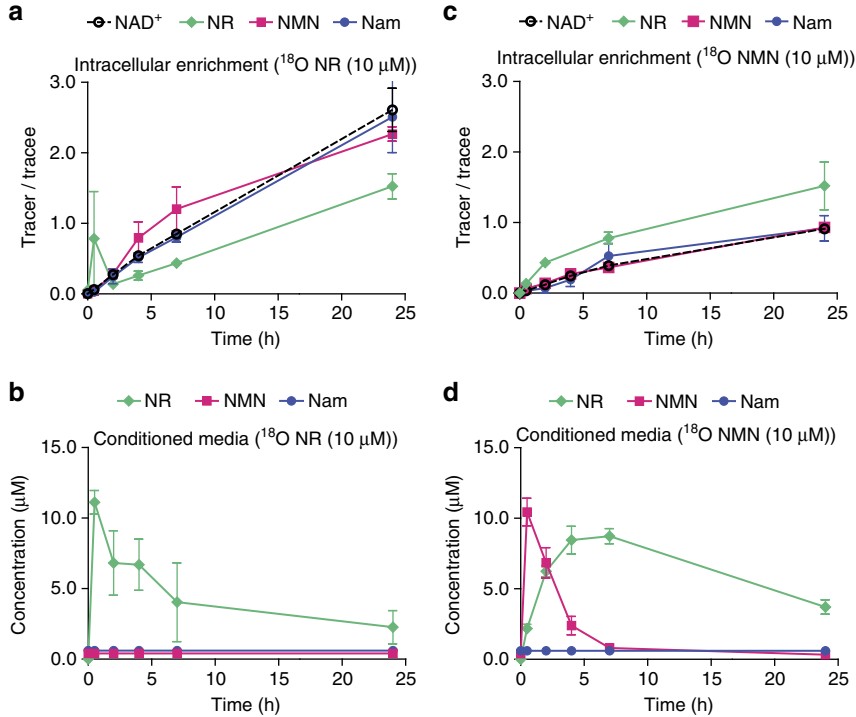

**Figure 4 | Extracellular NMN dephosphorylates to NR.** HepG2 cells were treated with [$^{18}$O]-labelled NR (**a,b**) or NMN (**c,d**). Intracellular enrichment of labelled compounds (**a,c**) and presence of labelled compounds in the media of treated cells (**b,d**). Results shown are mean ± s.e.m. ($n = 3$).

cultured in serum- and NAM-free medium. As shown in Fig. 4a, [$^{18}$O]-NR treatment of HepG2 cells led to a rapid rise in the ratio between heavy and endogenous NR, which was followed by incorporation of heavy label into NMN, NAD$^+$, and the product of NAD$^+$-consuming enzymes, NAM. This occurred in parallel with a drop in [$^{18}$O]-NR in the medium from 10 to 3 μM over the course of 24 h (Fig. 4b). No labelled [$^{18}$O]-NMN or [$^{18}$O]-NAM appeared during this time in the media of [$^{18}$O]-NR-treated HepG2 cells (Fig. 4b), indicating that NR is incorporated intact and serves as a direct NAD$^+$ precursor. In contrast, when cells were treated with [$^{18}$O]-NMN, there was a rapid appearance of extracellular [$^{18}$O]-NR (Fig. 4d) and enrichment of intracellular [$^{18}$O]-NR (Fig. 4c). Whereas more than half of the [$^{18}$O]-NR was internalized into NAD$^+$ metabolites at the 7 h time point, 80% of extracellular [$^{18}$O]-NMN was present as extracellular [$^{18}$O]-NR at this time point. The slow release of NR from NMN was accompanied intracellularly by a persistently high labelling of NR at all time points. These data are exactly consistent with extracellular NMN as a source of extracellular and intracellular NR and are kinetically inconsistent with transport of intact NMN through the plasma membrane.

**NRK1 is essential for NR and NMN to increase NAD$^+$.** We next aimed to determine if NRK1 is essential for NR and NMN metabolism into NAD$^+$ in liver-derived cells. Given the low transfection efficiency of HepG2 cells, we moved to AML12 hepatocytes, in which we could decrease *Nmrk1* mRNA and NRK1 protein levels by 60% using specific siRNAs (Fig. 5a). This knockdown was not sufficient to fully blunt NR-driven NAD$^+$ synthesis (Fig. 5b). These data suggested that residual NRK1 protein after the knockdown could account for the remaining NR-stimulated NAD$^+$ production. In line with this possibility, NR effectively increased NAD$^+$ in Hepa1.6 cells, which have significantly lower NRK1 levels compared with AML12 (Supplementary Fig. 2d,e).

To circumvent this problem, we generated a mouse model with a constitutive ablation of the *Nmrk1* gene (NRK1KO). Homozygous NRK1KO mice were born at expected Mendelian ratios and displayed no gross abnormalities. No *Nmrk1* mRNA or protein could be detected in any tissue examined including liver, kidney, brown adipose tissue or muscle, and no compensation of NRK2 in NRK1KO tissues was observed (Supplementary Fig. 3a,b). We then isolated primary hepatocytes from WT and NRK1KO mice. Hepatocytes isolated from WT mice showed comparable levels of NRK1 expression to that of endogenous liver (Supplementary Fig. 2e), whereas the NRK1 protein was undetectable in hepatocytes from NRK1KO mice (Fig. 5c). The expression of other enzymes related to NAD$^+$ biosynthesis (NAMPT and the NMNAT isozymes), NAD$^+$ consuming enzymes (CD38, CD157 and PARP-1) or ecto-5′-nucleotidases (CD73) was comparable between genotypes (Fig. 5c,d). Strikingly, NR and NMN increased NAD$^+$ levels in WT primary hepatocytes but not in primary hepatocytes from NRK1KO mice (Fig. 5e). In contrast, primary hepatocytes from NRK1KO mice increased NAD$^+$ levels on NA or NAM treatment normally, indicating a specific defect in NR utilization. Thus, NRK1 is required for the utilization of exogenous NR and NMN in liver cells.

Inability of NRK1-deficient hepatocytes to utilize NMN further suggests that NMN enters the cell in the form of NR. To further test this hypothesis, we blocked the possible NMN to NR conversion in hepatocytes using high concentration of cytidine monophosphate (CMP)[16]. Whereas CMP did not affect NR action, it fully inhibited the increase in NAD$^+$ on NMN supplementation, showing the necessity of NMN dephosphorylation to NR by an enzyme that is competitively inhibited by CMP (Fig. 6a).

To conduct the most stringent test of NMN utilization, we made use of wild-type and NRK1KO hepatocytes, double-labelled NAD$^+$ precursors, and a quantitative, targeted NAD$^+$ metabolomic flux experiment performed with LC-MS. NR and NMN were each synthesized such that the NAM ($^{13}$C) and

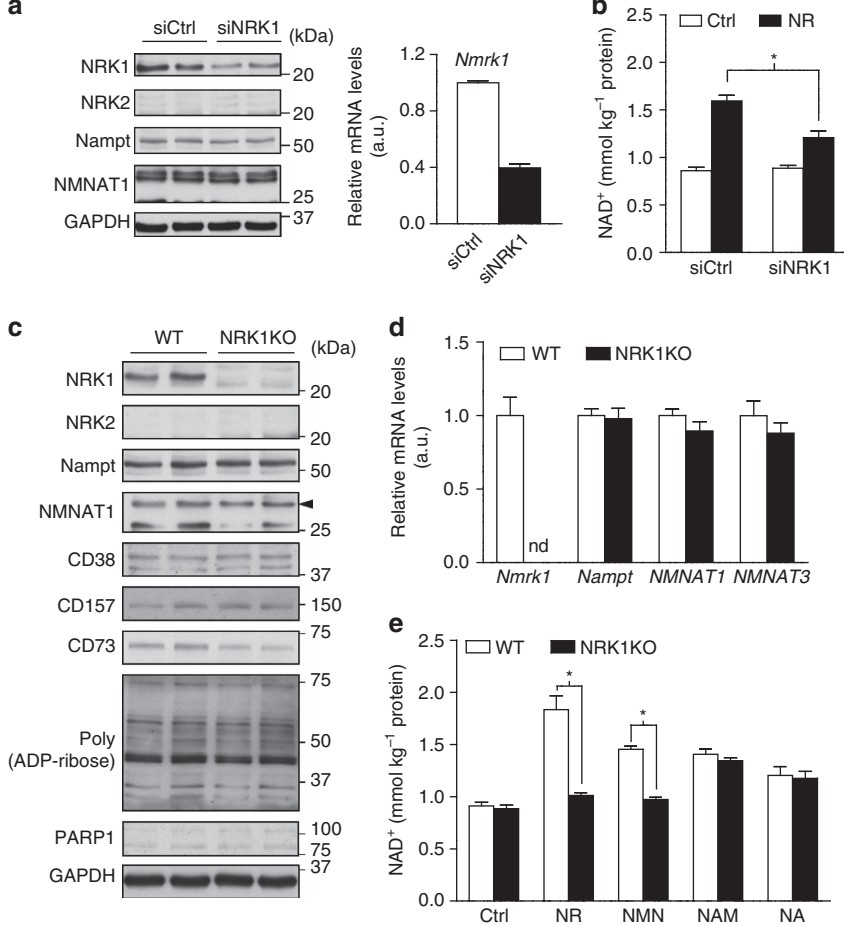

**Figure 5 | NRK1 is essential for NR and NMN to enhance NAD$^+$ synthesis in hepatic cells.** (**a**,**b**) AML12 cells were transfected with NRK1 siRNA. (**a**) Protein and mRNA levels of NRK1 and other NAD$^+$-related enzymes. (**b**) NAD$^+$ measurement after 6 h 0.5 mM NR treatment. *$P < 0.05$ versus ctrl by two-sided unpaired T-test. (**c**–**e**) Primary hepatocytes were isolated from 10 to 15 weeks old WT and NRK1KO male mice. Protein (**c**) and mRNA (**d**) levels of NRK1 and other enzymes involved in NAD$^+$ metabolism. (**e**) NAD$^+$ measurement after 6 h treatment with 0.5 mM NR, 0.5 mM NMN, 5 mM NAM, 0.5 mM NA or vehicle control. Results shown are mean ± s.e.m., *$P < 0.05$ versus WT by two-sided unpaired T-test. ($n = 3$).

the ribose ($^2$H) moieties are labelled. Primary hepatocytes from WT and NRK1KO mice were exposed to these compounds at 10 μM for 6 h and the extent of incorporation of double-labelled NR and NMN into double-labelled intracellular NAD$^+$ metabolites was determined. As shown in Fig. 6b, double-labelled NR has a profound advantage in incorporation of double-labelled NMN, NAD$^+$ and NADP with respect to double labelled NMN. By the 6 h time point, 40–50% of cellular NMN and NAD$^+$ and nearly 10% of cellular NADP incorporated both labels when wild-type hepatocytes were exposed to double-labelled NR. If, as previously reported[9], primary hepatocytes incorporate an intact NMN to form NAD$^+$, then exposure to double labelled NMN would have produced double-labelled intracellular NMN and potentially a faster and/or more highly incorporated population of intracellular metabolites. However, intracellular double labelled NMN was not observed in the 6 h time course nor were appreciable levels of any double-labelled NAD$^+$ metabolite observed when wild-type hepatocytes were exposed to double-labelled NMN (Fig. 6b). This would be consistent with a slower and/or less potent action of NMN due to the need for dephosphorylation into NR to drive NAD$^+$ synthesis. As shown in Fig. 6c, the ability of hepatocytes to convert double labelled NR to all double-labelled NAD$^+$ metabolites is completely NRK1-dependent. This experiment shows conclusively that NMN does not bypass

NRK1 and that liver cells depend on NRK1 to convert extracellular NR to NAD$^+$.

**NRK1KO mice are deficient in NR and NMN utilization in vivo.** Finally, to dissect the dependence of NAD$^+$ precursors on NRK1 in the in vivo synthesis of hepatic NAD$^+$, we delivered 500 mg kg$^{-1}$ NR, NMN, NAM or saline to wild-type and NRK1KO mice by intraperitoneal injection. All precursors led to large increases of NAD$^+$ levels in WT livers collected 1 h after injection (Fig. 7a). NR increased liver NAD$^+$ content by ∼220% in WT mice (Fig. 7a,b), whereas NAM and NMN increased NAD$^+$ over basal levels to a lesser extent (∼170%). While the effects of NAM were similar in WT and NRK1KO mice, the responses to both NR and NMN were significantly blunted in NRK1KO mice compared to WT controls (Fig. 7a,b). However, about 60% of the NR and NMN effect on NAD$^+$ remained in NRK1KO mice. To rule out the influence of nonspecific conversions due to high doses of compounds used, we also injected mice with lower doses of 50 mg kg$^{-1}$. At the lower doses, NRK1 deficiency only partially blunted the increase in hepatic NAD$^+$ production after NR and NMN injection (Supplementary Fig. 3c).

In addition to liver, we evaluated kidney, BAT and skeletal muscle NAD$^+$ levels after IP delivery of the different

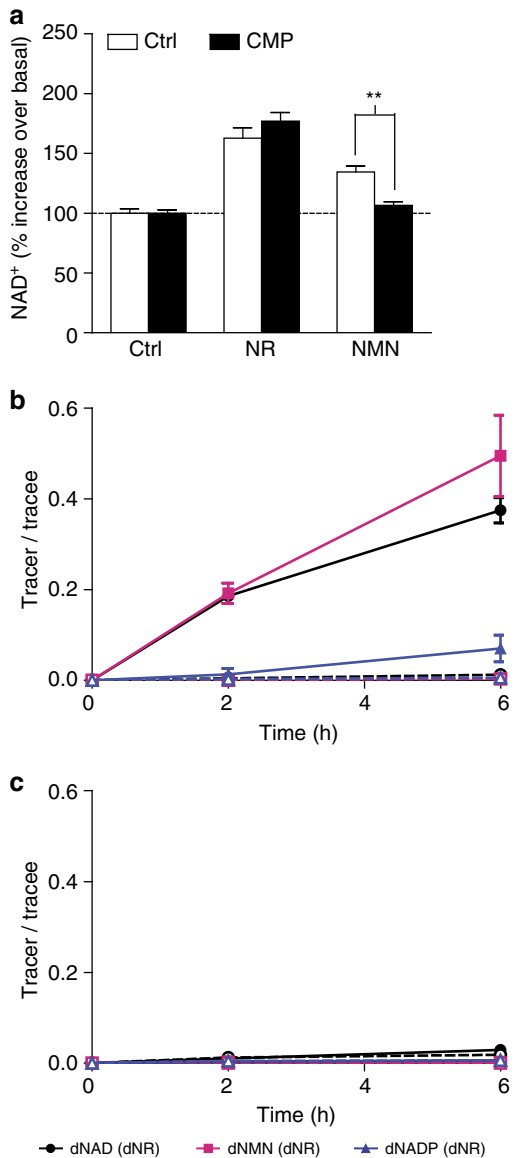

**Figure 6 | Extracellular conversion of NMN to NR by hepatocytes.** (**a**) Primary hepatocytes were isolated from 10 to 15 weeks old WT mice. Cells were treated with $2 \mu M$ FK866 and $\pm 2 mM$ CMP for 1 h. Then, 0.5 mM NR or NMN was added and $NAD^+$ was measured after 2 h. (**b,c**) WT (**b**) and NRK1KO (**c**) hepatocytes were treated with $[^2H]$-$[^{13}C]$-double labelled NR or NMN. Intracellular enrichment of labelled compounds. Results shown are mean $\pm$ s.e.m., **$P < 0.01$ versus ctrl by two-sided unpaired $T$-test ($n = 3$).

$NAD^+$ precursors. As in liver, the ability of NAM to trigger $NAD^+$ synthesis was comparable between WT and NRK1KO mice in all tissues analysed (Fig. 7c–h). In line with high NRK1 expression in kidney (Fig. 3a and Supplementary Fig. 1a), NR and NMN were very effective $NAD^+$ precursors in this tissue, and their ability to enhance $NAD^+$ synthesis was blunted by $\sim 70\%$ in NRK1 deficient mice (Fig. 7c,d). Opposite to kidney, BAT has rather moderate NRK1 expression and relatively lower increases in $NAD^+$ levels, but NR and NMN actions were also largely blunted in NRK1 KO mice (Fig. 7e,f). These results indicate that in liver, kidney and BAT, NR and NMN actions on $NAD^+$ synthesis after IP delivery are largely, but not completely, blunted by NRK1 deficiency. Muscle was the tissue with the lowest response to $NAD^+$ precursors, which might explain why

we could not detect major differences between WT and NRK1KO mice in their response to NR or NMN. Taking into account that in muscle there is high expression of NRK2, we hypothesized that NRK2 might be able to compensate for NRK1 deficiency. We therefore intraperitoneally administered NAM, NR and NMN to *Nmrk1* and *Nmrk2* double KO mice (NRK1/NRK2DKO). However, WT mice and NRK1/NRK2DKO displayed comparable increases in $NAD^+$ levels in skeletal muscle after intraperitoneal NR and NMN administration (Supplementary Fig. 3d).

In yeast, an NRK-independent, NR salvage pathway is initiated by the activities of Urh1, Pnp1, and, to a slight degree, Meu1, which split NR into a ribosyl product and NAM[26]. Because Pnp1 is the yeast homologue of mammalian purine nucleoside phosphorylase[29], we anticipated that NAM produced by breakdown of intraperitoneal injected precursors could be responsible for NRK1-independent synthesis of $NAD^+$ in liver.

We therefore analysed NR, NMN and NAM levels in plasma and livers obtained from these animals. Injection of NAM did not significantly alter hepatic NAM content, but led to comparable increases in NAM plasma levels ($\sim 45$-fold) in both WT and NRK1KO mice (Fig. 8a,b, respectively). Strikingly, both NR and NMN injection also led to significant increases of NAM in plasma, independent of genotype (Fig. 8b). This indicates that both NR and NMN are partially converted to NAM upon intraperitoneal injection, thereby accounting for the ability of NRK1KO mice to increase $NAD^+$ content in liver upon NR or NMN administration. To test whether the breakdown of NR and NMN into NAM is due to chemical or enzymatic degradation, we determined the stability of these compounds in isolated murine plasma. NR incubation in murine plasma leads to relatively quick degradation, with $\sim 10\%$ of NR degraded after 10 min and $\sim 66\%$ degraded after 1 h (Fig. 8e), which is further illustrated by gradual increase in NAM abundance in the samples (Fig. 8g). On the contrary, NMN is stable in plasma and there is no NAM increase in NMN samples up to 1 h incubation (Fig. 8f,g). The results indicate the existence of plasma factor that degrades NR into NAM. Moreover, it additionally hints that the appearance of NAM in plasma of mice injected with NMN may come after initial conversion of NMN into NR. Importantly, NR is stably associated with protein fractions in milk[25] with a lifetime of weeks and may be circulated in a cell-associated form in animals with a lifetime of hours. NR and NMN are both stable when incubated in minimal essential medium (MEM) at 37 °C up to 6 h (Supplementary Fig. 4a,b). However, NR but not NMN degradation with appearance of NAM could be observed when supplementing media with 10% FBS (Supplementary Fig. 4a–c). Therefore, the increases in $NAD^+$ observed in NRK1KO tissues after NMN or NR administration likely derive from the conversion of these precursors to NAM.

In our analyses, NMN levels were poorly detectable in mouse plasma, at concentrations around the detection limit of our assays ($\sim 20 nM$). Though plasma NR levels were also at the limit of detection, NR-injected NRK1KO mice displayed a significant increase in circulating NR levels compared with vehicle treatment (Fig. 8d). Further, hepatic NR increased in NRK1KO mice after administration of NR and modestly after NMN (Fig. 8c), establishing that NR entry into the liver is unaffected in NRK1KO mice but accumulates due to low NR phosphorylation. The data confirm that NRK1 is crucial for maximal $NAD^+$ elevation in response to both NR and NMN.

**Discussion**

Supplementation with NR and NMN has recently been shown to promote largely overlapping benefits on metabolic health,

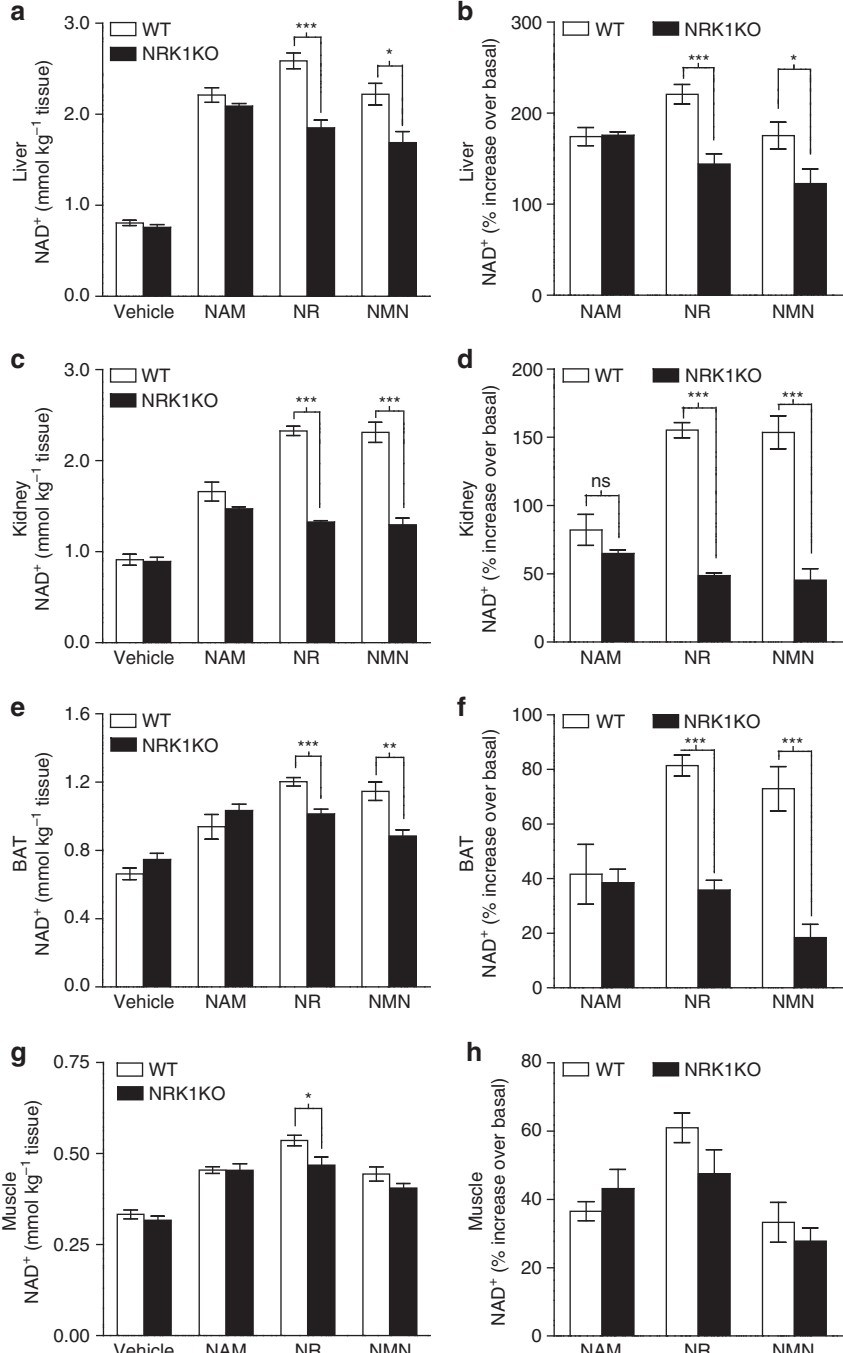

**Figure 7 | NAD$^+$ *in vivo* metabolism in NRK1KO mice.** Five to six weeks old WT and NRK1KO male mice were IP injected with 500 mg kg$^{-1}$ of the indicated compounds. Then, liver, kidney, brown adipose (BAT) and skeletal muscle samples were collected after 1 h, $n = 5$ per group. NAD$^+$ measurement in liver (**a**), total kidney extracts (**c**), BAT (**e**) and muscle (**g**) tissue. Percentage increase above basal NAD$^+$ level in liver (**b**), kidney (**d**), BAT (**f**) and muscle (**h**) tissue upon treatments. Results shown are mean ± s.e.m., \*$P < 0.05$, \*\*$P < 0.01$, \*\*\*$P < 0.001$ versus WT by two-sided unpaired *T*-test ($n = 5$).

mitochondrial and neurodegenerative disorders[3]. Although both compounds lead to NAD$^+$ synthesis, specific analysis of their bioavailability and metabolism remained to be elucidated. Our work identifies NRK1 as the central rate-limiting enzyme for the utilization of both compounds and, consequently, for their metabolic benefits.

NRK1 and NRK2 were initially described as highly conserved mammalian homologs of yeast Nrk1 with the ability to convert NR to NMN in a newly identified pathway of eukaryotic NAD$^+$ biosynthesis[18]. Despite similar enzymatic activities, these enzymes display distinct tissue expression patterns. The baseline

expression of the *Nmrk2* gene is highly restricted to muscle and this enzyme proved to be unstable when ectopically expressed in cultured fibroblasts. This led us to focus on the more ubiquitously expressed NRK1 isozyme. Here we show that although NRK1 is dispensable for maintenance of basal NAD$^+$ levels in cells cultured in standard NAM-containing media, it is rate-limiting for cells to utilize NR and NMN as sources of NAD$^+$. In sharp contrast, NRK1 is dispensable for conversion of NAM and NA into NAD$^+$. In addition, our data from quantitative mass spectrometry clearly show that *in vitro* NR supplementation does not lead to NAM accumulation in the media at any point before

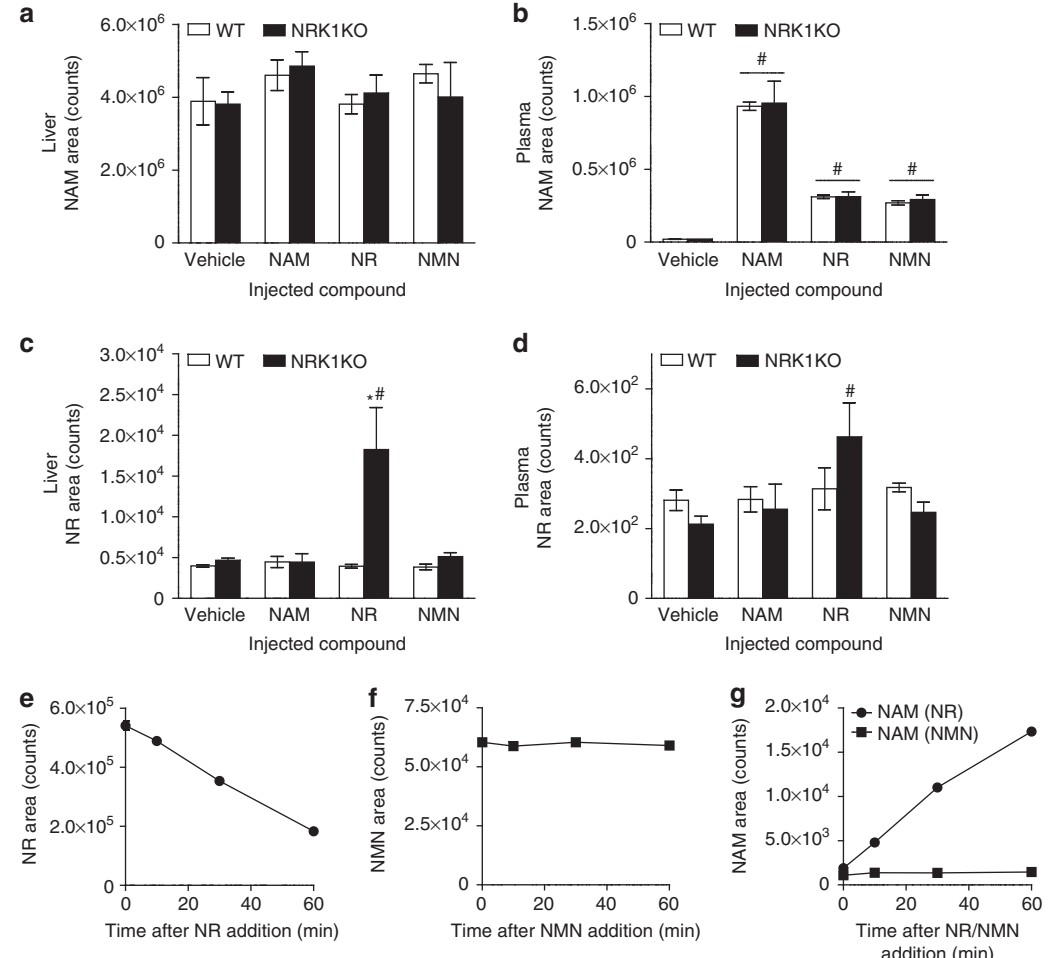

**Figure 8 | Intraperitoneal injected NR and NMN partially degrades to NAM.** (**a–d**) Five to six weeks old WT and NRK1KO male mice were IP injected with 50 mg kg$^{-1}$ of indicated compound and blood liver samples were collected after 1h, $n = 4$ per group. Then, NAM level in liver (**a**) and plasma (**b**) were measured, as well as NR level in liver (**c**) and plasma (**d**) of these animals. (**e–g**) NR and NMN stability in isolated murine plasma. Plasma samples were isolated from NRK1KO mice. 0.5 mM NR or NMN was incubated with plasma at 37 °C for 60 min. NR (**e**), NMN (**f**) and NAM (**g**) level was measured in samples by LC-QqQ MS. Results shown are mean ± s.e.m., *$P < 0.05$ versus WT; #$P < 0.05$ versus vehicle by one-way ANOVA followed by Bonferroni's *post-hoc* test ($n = 3$).

NAD$^{+}$ synthesis. *In vivo*, however, intraperitoneal NR and NMN administration quickly lead to increases in NAM. Our results further demonstrate that while NR is spontaneously converted to NAM in cell-free plasma, NMN is more resistant to this process *in vitro*. This indicates that the comparable increases in NAM levels observed after NMN administration likely derive from NMN to NR dephosphorylation. The chronic effects of NR or NMN on health, however, are unlikely to rely on their transformation to NAM, for example, many of the benefits of NR and NMN are attributed to sirtuin activation, as both compounds led to higher sirtuin activities from yeast to mammals[9,19–21,23,28,30]. NAM, in contrast, is a well reported and widely used sirtuin inhibitor[31] and has been recently shown to inhibit NAD$^{+}$-consuming activities *in vivo*[24]. Similarly, both NMN and NR protect against insulin resistance and axotomy-induced axonal degeneration[9,19,32], an effect that has not been observed with NAM[32,33].

While extracellular dephosphorylation of NMN into NR is necessary for its utilization in yeast[34], the means by which NMN is utilized by mammalian cells remained controversial. NMN was proposed to circulate at high abundance and serve as a whole body NAD$^{+}$ precursor[9,14]. However, NMN has proven challenging to detect in blood. Whereas NMN

concentration has been reported to reach around 50 µM in plasma[14], our study and others[13] have failed to detect comparable levels of circulating NMN. These differences might be consequent to the use of different methodologies to measure NMN. For example, using one-dimensional HPLC-based methods, NMN intracellular concentrations were reported to rise up to ~400–500 pmol mg$^{-1}$ of white adipose and pancreatic tissue 15 min after intraperitoneal injection of 500 mg kg$^{-1}$ of NMN, parallel to a peak level of NAD$^{+}$ in white adipose tissue of ~50 pmol mg$^{-1}$ (ref. 9). However, by LC-tandem MS and with the use of internal standards, the baseline concentrations of hepatic NMN and NAD$^{+}$ were determined to be ~2 and ~1,000 pmol mg$^{-1}$, respectively, rising to ~10 and ~4,000 pmol mg$^{-1}$, respectively, 6 h after oral gavage of 185 mg kg$^{-1}$ NR[24]. This is also consistent with recent reports of NMN levels around 1.5 pmol mg$^{-1}$ tissue in tumours and ~80 nM NMN in ascites fluid[17]. These results highlight the need to standardize NAD$^{+}$-metabolomic measurements with methods that are quantitative and multidimensional.

Irrespective of the circulating NMN concentration, our work indicates that NMN dephosphorylation to NR constitutes a critical step in order to act as an exogenous NAD$^{+}$ precursor. This is sustained by several studies suggesting that the extracellular

receptor CD73 could act as a potential NR-releasing enzyme[15]. CD73 possesses pyrophosphatase and 5′-ectonucleotidase activities that enable the utilization of extracellular $NAD^+$ and NMN by converting them to NR, which would then enter the cell for $NAD^+$ synthesis[15]. In line with this, CD73 silencing blocks the use of NMN as an extracellular $NAD^+$ precursor[17]. Our NRK1 gain- and loss-of-function models provide further evidence that NMN is extracellularly dephosphorylated to NR for its utilization as an $NAD^+$ precursor. If NMN were directly incorporated into the cell, its conversion to $NAD^+$ would only require NMNAT and not depend upon NRK1 activity. These data, together with our data on CMP inhibition and kinetic studies using tracer compounds, provide compelling evidence for the extracellular conversion of NMN to NR for subsequent conversion to $NAD^+$ through the NRK pathway. Though yeast cells express a dedicated NR transporter[35], most of the evidence suggests that mammalian cells rely upon equilibrative nucleoside transporters (ENTs) to import NR[16,17]. Inhibition of ENTs also largely prevented NR-induced $NAD^+$ synthesis in our F3T3-NRK1 cells (Supplementary Fig. 5a). Accordingly, NMN was unable to generate $NAD^+$ when ENTs where pharmacologically blocked[16,17], indicating that NMN extracellular dephosphorylation is not a side reaction occurring concomitant with NMN direct transport but is the exclusive pathway for utilization of extracellular NMN in $NAD^+$ synthesis.

NRK isozymes are highly conserved through evolution, which indicates a primordial need for NR as an $NAD^+$ precursor[18]. In addition, the induction of NRK2 in response to cellular damage or stress[32] suggests the existence of a demand for circulating NR. The natural source of oral NR, NAM and NA is largely the $NAD^+$ in the food we eat as broken down by digestion and the microbiome[2,3]. Interestingly, it was recently shown that human cytosolic 5′-nucleotidases can catalyse the conversion of NMN into NR inside the cell and that at least some cells are able to release nucleoside precursors of $NAD^+$ synthesis for other cells[36], which also suggests that NR is in circulation. Finally, the presence of NR in bodily fluids is also supported by the fact that several bacteria affecting humans, such as *Haemophilus influenza* and *Streptococcus pneumoniae* rely on exogenous NR to support $NAD^+$ synthesis and life[37,38]. Future work will be required to fully characterize circulating $NAD^+$, NMN and NR, which may be cell-associated. Nonetheless, the different efficacy of NAM and NR to boost $NAD^+$ synthesis and protect against metabolic dysfunctions, indicate that NR and NAM are distinct vitamins with non-identical tissue and cellular activities. This concept was established by a recent study in which orally administered equimolar NAM, NR and NA produced unique effects on hepatic $NAD^+$ metabolism through distinct biosynthetic routes[24]. This experiment, conducted in parallel with a human clinical trial at the same effective dose of NR, showed that NR produces the greatest rise in liver $NAD^+$ synthesis, followed by NAM and NA[24].

By establishing NRK1 as a rate-limiting and essential enzyme for NR and NMN metabolism, we highlight NR and NMN as convergent NR supplementation strategies. Further work will be needed to optimize compositions or formulations that may be preferentially targeted to particular organs to provide nutritional and therapeutic benefits.

## Methods

**Materials.** All chemicals and reagents were purchased from Sigma-Aldrich unless specified otherwise. NR Cl salt was synthesized and provided by Biosynth. $^{18}O$-labelled NR[39] was synthesized from $^{18}O$-nicotinamide, prepared by hydrolysis of cyanopyridine in $^{18}O$-water. $^{18}O$-labelled NMN was prepared from $^{18}O$-NR by phosphorylation with NRK1 (ref. 18). $^{13}C$-nicotinamide was synthesized from $^{13}C$-nicotinic acid (purchased from Toronto Chemical Research) according to Von Elverfeldt et al.[40]. Labelled D-ribofuranose 1,2-$^2H$,3,5-tetraacetate (precursor

to Vorbrüggen glycosylation) was synthesized from D-[2-$^2H$]-ribose (purchased from Omicron Biochemicals) according to Begley et al.[41] Double labelled $^{13}C$-$^2H$ nicotinamide riboside was synthesized from $^{13}C$-nicotinamide and D-ribofuranose 1,2-$^2H$,3,5-tetraacetate according to Vorbrüggen's methodology described by Sobol et al.[42]. $^{13}C$-$^2H$ nicotinamide mononucleotide was synthesized from double labelled $^{13}C$-$^2H$ nicotinamide riboside according to procedures described by Lee et al. and references therein[43]. Analytical methods and reports on labelled and unlabelled NR and NMN are provided as Supplementary Methods and Supplementary Fig. 6, respectively. NMN was purified by HPLC on a strong anion exchange column with a 10–750 mM gradient of $KH_2PO_4$ (pH 2.6). Isotopic purity was assessed by LC-MS and concentration was assessed using an extinction coefficient (260 nm) of $4,200\,M^{-1}\,cm^{-1}$.

**Plasmids.** The coding regions of mouse *Nmrk1* (NM_145497.2) and *Nmrk2* (NM_027120.2) were amplified from Origene clones MC204637 and MC210962, respectively, using KOD polymerase (Merck). The resulting PCR products was digested with *Bamh1-Not1* or *Bgl2-Not1* and ligated into a Flag modified pCMV5 vector (AF239249.1), in which the *Not1* site was in frame with the coding regions to provide a C-terminal Flag tag on the proteins. In addition, forms with point mutations (D36A and E98A in NRK1 as well as D35A and E100A in NRK2) were created by PCR site-directed mutagenesis to generate enzymatically inactive versions[26]. Sequences of all the clones were verified using the BigDye Terminator 3.1 kit and 3500XL Genetic analyzer (Applied Biosystems).

**Cell culture.** All the cell lines used in the study come from ATCC and have been tested for mycoplasma contamination using MycoProbe Mycoplasma Detection Kit (R&D cat. CUL001B). NIH/3T3, Hepa1–6 and HepG2 cells were cultured in DMEM supplemented with 10% FBS. AML12 cells were cultured in DMEM/F-12 medium supplemented with 10% FBS, 1x ITS-G and 100 nM dexamethasone. C2C12 cells were cultured in DMEM supplemented with 20% FBS and differentiated in DMEM supplemented with 2% horse serum for 4 days. For overexpression experiments, cells were transfected with 4 μg of plasmid DNA using Lipofectamine 3000 (Thermo Fisher) according to manufacturer's instructions and the treatments were performed 48 h post transfection. For knockdown experiments, AML12 cells were transfected with 75 pmol siRNA (siRNA against mouse *Nmrk1* was obtained from Thermo Fischer) using Lipofectamine RNAiMAX (Thermo Fisher) according to manufacturer's instructions and the treatments were performed 48 h post-transfection. NRK1 or NRK2 overexpressing stable cell lines were generated from NIH/3T3 cells using FlpIn system (Invitrogen), in which an expression vector containing the *Nmrk1* or *Nmrk2* gene was integrated into the genome via Flp recombinase-mediated DNA recombination at the FRT site. FlpIn cells were cultured in DMEM supplemented with 10% FBS and 100 μg ml$^{-1}$ Hygromycin B. For $NAD^+$ precursor response measurements, cells were treated with 0.5 mM NR, 0.5 mM NMN, 0.5 mM NA or 5 mM NAM for 6 h unless specified otherwise.

**Nmrk1 KO mouse generation.** NRK1-deficient mouse models were generated on a pure C57BL/6NTac background at Taconic Biosciences. Briefly, exons 3 to 7 of *Nmrk1* gene were flanked with loxP sites and floxed mice were crossed with mice expressing the Cre recombinase under the general promoter of the Gt(ROSA)26Sor gene (Cre deleter). The deletion of these exons was validated by PCR (Supplementary Fig. 8), and resulted in the complete loss of the NRK1 protein. Mice carrying the whole body, including germ line, *Nmrk1* deletion were further bred to eliminate expression of the Cre recombinase. Age- and body weight-matched males were randomly assigned into different treatments. Samples from animal interventions were analysed and quantified in a blind manner.

**Animal care.** Unless otherwise specified, mice were kept in a standard temperature- and humidity-controlled environment with a 12:12-h light:dark cycle. Mice had nesting material and *ad libitum* access to water and a low-fat diet (D12450J, from Research Diets Inc.). All animal experiments were carried according to Swiss and EU ethical guidelines and approved by the local animal experimentation committee under license 2770.

**Primary hepatocyte isolation.** Hepatocytes were isolated WT and NRK1KO mice by continuous recirculating perfusion of the mouse liver *in situ* with collagenase digestion[44]. Perfusion was performed in Krebs buffer (4.7 mM KCl, 0.7 mM $KH_2PO_5$, 10 mM HEPES, 117 mM NaCl, 24.6 mM $NaHCO_3$, 0.2% glucose) supplemented with 5 mM $CaCl_2$ and 0.5 mg ml$^{-1}$ collagenase (Worthington, type IV) for 10 min with 5 ml min$^{-1}$ flow. Cells were seeded in M199 containing 100 U ml$^{-1}$ penicillin G, 100 μg ml$^{-1}$ streptomycin, 0.1% (w/v) BSA, 10% (v/v) FBS, 10 nM insulin, 200 nM triiodothyronine and 100 nM dexamethasone. Post attachment (4–5 h), cells were cultured overnight in M199 supplemented with antibiotics and 100 nM dexamethasone and used for experiments the following morning.

**Primary BAT cells.** Primary brown adipocytes cells were obtained from the interscapular BAT of WT mice. Immortalized brown pre-adipocytes were grown

until 90% confluence in 'growth medium' (DMEM supplemented with 10% FBS, 0.02 μM Insulin, and 1.5 nM 3,3',5 Triiodothyronine (T3)) and next differentiated during 36 h with growth medium supplemented with 0.5 μM dexamethasone, 1 μM rosiglitazone, 0.125 μM indomethacin and 0.5 mM isobutylmethylxanthine. Then, cells were cultivated in growth medium until day 6 of differentiation[45].

**Western blotting.** Cells were lysed in 50 mM Tris-Cl pH 7.5, 150 mM NaCl, 5 mM EDTA, 1% NP40, 1 mM Na butyrate with protease inhibitors. Proteins were quantified using a BCA assay (Pierce). For western blotting, proteins were separated by SDS– polyacrylamide gel electrophoresis and transferred onto nitrocellulose membranes before incubation with primary antibodies against Flag M2 F1804, α-tubulin T9026 (Sigma); NAMPT A300-372A (Bethyl); CD73 #13160, GAPDH #2118 (Cell Signaling); CD38 sc-7049, CD157 sc-7115, PARP1 sc-1561 (SantaCruz); poly(ADP-ribose) ALX-210–890 (Enzo); or NRK1 and NRK2. NRK1 and NRK2 antibodies were purified by YenZym Antibodies from rabbits immunized with CTRSEEDLFSQVYEDVKQELEKQNGL and CKSPEGLFHQV-LEDIQNRLLNTS peptides, respectively. Primary antibodies were used in 1:5,000 dilution for GAPDH and α-tubulin and 1:1,000 dilution for others. Antibody detection reactions were developed by enhanced chemiluminescence (Amersham). All uncropped western blots are available in Supplementary Fig. 9.

**RNA extraction and qPCR.** Total mRNA from all studied tissues or cells was extracted using TRIzol (Life Technologies) according to manufacturer's instructions. RNA concentrations were measured with Nanodrop 1000 (Thermo Scientific). Reverse transcription was performed using SuperScript II (Life Technologies) with oligo dT plus random hexamer primers and RNAsine (Roche) according to the manufacturer's protocol. Quantification of mRNA expression was performed using SYBR Green real time PCR technology (Roche). Reactions were performed in duplicate in a 384-well plate using the Light Cycler (Roche). Gene expression was normalized with *b2-microglobulin* and *cyclophillin* as housekeeping genes. Relative gene expression between genotypes was assessed using the ΔΔCt method. Primers used: *Nmrk1* (5′-CCCAACTGCAGCGTCATA TC-3′), *Nmrk2* (5′-GCCGTATGAGGAATGCAAGC-3′), *Nampt* (5′-AGTGGCCA CAAATTCCAGAGA-3′), *Nmnat1* (5′-TGGCTCTTTTAACCCCATCAC-3′), *Nmnat3* (5′-TCACCCGTCAATGACAGCTAT-3′), *b2-microglobulin* (5′-ATGGGA AGCCGAACATACTG-3′), *cyclophillin* (5′-CAGGGGAGATGGCACAGGAG-3′).

**NAD$^+$ assay.** NAD$^+$ was extracted from cells or tissues and measured by EnzyChrom NAD/NADH Assay Kit (BioAssay Systems) performed according to manufacturer's instructions.

**NAD$^+$ metabolome enrichment analysis using labelled compounds.** NAD$^+$ metabolite enrichment in HepG2 cells, primary hepatocytes and cultured media was determined using LC-MS/MS[46]. Briefly, cells were incubated in NAM-free, serum-free DMEM for 24 h at which time cells were dosed with either labelled NR or labelled NMN at 10 μM final concentration. At 0, 0.5, 2, 4, 7 and 24 h, media were collected and snap-frozen while cells were trypsinized, centrifuged, washed with ice-cold PBS and snap-frozen in liquid nitrogen. All media and cells were stored at − 80 °C before analysis. Media were diluted 1:1 with LC-MS grade water containing internal standard (10 μM cytidine) and injected for analysis. Cell pellets were washed in ice-chilled PBS, resuspended in 300 μl of a 75% ethanol/25% 10 mM HEPES, pH 7.1 solution that had been preheated to 80 °C. Samples were then shaken at 1,000 r.p.m. at 80 °C for 3 min. Soluble metabolites were recovered after 10 min centrifugation at 16 kg (ref. 46). A260 values of extracts were determined using a Thermo Scientific 2000c Nanodrop. All cellular extracts were diluted to an A260 value of 7, injected and analysed. Extracellular metabolites were quantified by comparing the ratio of analyte area to the area of internal standard in sample compared with a standard curve in water. Intracellular enrichment was measured by dividing the $^{18}$O analyte areas by their $^{16}$O analyte isotopologues. Enrichments were corrected for natural isotope abundance where necessary. Further details on the synthesis of labelled compounds and multiple ion detection spectra of unlabelled and double labelled NR and NMN are described in Supplementary Figs 6 and 7.

**Analysis of NR, NAM and NMN by LC-QqQ MS.** Fifty microlitres of freshly thawed plasma and 2 mg of lyophilized liver were mixed with 150 and 200 μl, respectively, of 100 mM ammonium acetate water:acetonitrile 15:85 (v/v). Plasma extract was vortexed vigorously for 20 s, while liver extract was homogenized with 0.5 mm stainless steel beads using a Bullet Blender (Next Advance). The resulting plasma and liver extracts were incubated on ice for 1 h for protein precipitation and centrifuged at 15,000 r.p.m. at 4 °C for 10 min. The resulting supernatant was transferred to an LC-MS vial. Plasma extract supernatant was diluted 1:10 in 100 mM ammonium acetate water:acetonitrile 15:85 (v/v), while liver extract supernatant was directly analysed by LC-MS. To avoid degradation effects analyses were done within 12 h.

NAM, NR and NMN metabolites were determined in a 1290 UHPLC coupled to a triple quadrupole 6490 iFunnel QqQ/MS (Agilent Technologies), using an Acquity UPLC BEH HILIC column of 1.7 μm and 2.1 × 100 mm (Waters). Injected

sample volume was 1 μl at a flow rate of 0.5 ml min$^{-1}$ using 100 mM ammonium acetate (solvent A) and acetonitrile (solvent B) as mobile phases with an elution gradient of 0–1 min 85% B, 2 min 20% B, 2–3 min 20% B isocratic, 3–3.5 min 15% B, 3.5–7 min 15% B isocratic and 8 min 85% B. ESI conditions were 150 °C and 12 l min$^{-1}$ for drying gas temperature and flow, 20 p.s.i. of nebulizer gas pressure, 350 °C and 12 l min$^{-1}$ for sheath gas temperature and pressure, 3,500 V of capillary voltage and 500 V of nozzle voltage. The QqQ was operated in multiple reaction monitoring mode and positive polarity, applying a fragmentor voltage of 380 V and a cell accelerator voltage of 3 V. Multiple reaction monitoring transitions were: NR (255→123, 106), NAM (123→78, 80, 53), NMN (335→123, 97, 80). Quality control (QC) samples consisting of pooled plasma or liver extracts, were alternatively inserted along the run sequence. Furthermore, experimental samples were randomized to reduce systematic error associated with instrumental drift. Relative quantification of compounds was done by comparison of the integrated areas (Supplementary Table 1).

**Statistical analyses.** Statistical analyses were performed with GraphPad Prism version 5.02 for Windows (La Jolla, CA, USA). *In vitro* analysis were performed in duplicates in three separate experiments. On the basis of an a priori power analysis, the studies were powered to allow significant detection of differences reaching medium size (20–30%) with >80% power in case the number of estimated false positive is set to 5%. Differences between two groups were analysed using a Student's two-tailed *t*-test. One-way ANOVA analysis with Bonferroni post-test was used when comparing more groups. Group variances were similar in all cases. A $P$ value <0.05 was considered significant. Data are expressed as means ± s.e.m.

**Data availability.** The data that support the findings of this study are available from the corresponding authors upon request.

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

## Acknowledgements

We thank members of the Canto, Migaud and Brenner laboratories for helpful discussions, José-Luis Sanchez-Garcia, Roger Hunter, Serge Ducommun and Kei Sakamoto for help with primary hepatocyte isolations, Maria Deak for help with plasmids generation, and the members of the EPFL and University of Iowa animal facilities for technical support. We thank Advanced ASSET centre, Institute for Global Food Security and Dr O. Chevallier for performing HRMS analysis. This study was supported by the Nestlé Institute of Health Sciences S.A., grants from the Roy J. Carver Trust and National Institutes of Health (R21-AA022371) to C.B. and from the Biotechnology & Biological Sciences Research Council (BB/N001842/1) to M.E.M.

## Author contributions

J.R., S.A.J.T., J.A., C.B. and C.C. designed the study. J.R., M.J., M.B. and S.S.K. performed the experiments. S.A.J.T., M.R., P.R., M.M. and C.B. performed tracer experiments. R.R., N.C. and O.Y. performed LC-QqQ MS analyses of NAD + precursors. All authors analysed data. J.R., C.B. and C.C. wrote the manuscript and all authors edited and approved the final manuscript.

## Additional information

**Competing financial interests:** J.R., M.J., M.B., S.S.K. and C.C. are employees of the Nestlé Institute of Health Sciences S.A. C.B. owns stock in ChromaDex and has received a research grant and serves on the scientific advisory board of ChromaDex. C.B. is co-founder and Chief Scientific Adviser of ProHealthspan, which distributes an NR supplement. M.E.M. has received research grants and serves as a consultant for ChromaDex.

