## [Peer review file · Nature Communications]

Reviewers' comments:

Reviewer #1 (Expert in metabolomics; Remarks to the Author):

In this paper the authors have used stable isotopes and mass spectrometry to identify the metabolism of nicotinamide mononucleotide in vitro. Mass spectrometry was also used to analyze metabolites in mouse plasma and liver extracts. The analytical methodologies for metabolite analysis and quantification are robust but there are a few minor comments that should be addressed:

-The method of cell harvesting could potentially affect cellular metabolism. It has been previously shown that trypsin affects central carbon metabolites (Badur, MG, Biotechnol. J. 2015, Scoazec, M, Methods in Enzymology, 2014).

-For Qqq analysis, was there an internal standard present in the samples to account for analytical variability?

-It would be useful to see the MID plots for the isotopologues formed from [18O]-NR and [18O]-NMN.

-It would also be good to have a schematic of the metabolic pathways.

Reviewer #2 (Expert in NAD metabolism; Remarks to the Author):

A. Summary of the key results

Using cell lines, primary cell culture and mouse model, this manuscript showed that NR is a more effective NAD⁺ precursor relative to niacin (NA, NAM) and that Nrk1 (NR kinase) plays a key role in NR metabolism under defined conditions. Interestingly, this study also showed that NMN, another NAD⁺ precursor/intermediate, is extracellularly converted to NR, which is then transported into cell and converted to NAD⁺. NMN has been used in a number of studies to boost NAD⁺ levels in various cell/animal models. However, it remains unclear how extracellular NMN becomes intracellular NAD⁺. One possibility is that NMN may be broken down into smaller molecules such as NR or NAM. Overall this is an interesting study. Although experimental designs were mostly logical, the reviewer has a few concerns and suggestions.

B. Originality and interest: if not novel, please give references:

The novelty of this study lies in the development of Nrk1 knockout mouse model for the study of NR metabolism.

C. Data & methodology: Most approaches were valid. Quality of data and presentation were adequate.

D. Appropriate use of statistics and treatment of uncertainties: Statistical analyses were adequate.

E. Conclusions: robustness, validity, reliability: additional studies are needed to strengthen the major conclusions of this study.

F. Suggested improvements: see reviewer's suggestions listed below.

G. References: appropriate credit to previous work? Yes.

H. Clarity and context: lucidity of abstract/summary. Appropriateness of abstract, introduction and conclusions: Abstract and Conclusions need to be modified to reflect the fact that some

observations were obtained under specific conditions.

Reviewer's suggestions:

1. The title sentence is overstated. Nr1 is important but not essential for NR or NMN metabolism. As shown in Fig. 6a-6d, nr1KO liver and kidney cells were still able to convert a significant fraction of NR and NMN to NAD⁺. In addition, neither Nr1 over-expression nor nr1KO affects the basal level of NAD⁺ (Fig. 2d, 3d, 6a-d). The effects of Nr1 (either KO or overexpression) on NR or NMN utilization were only seen when cells (or mice) were treated with NR or NMN. Therefore, these results mainly justified the role of Nr1 in the utilization of supplemented NR/NMN. The role of Nr1 in NAD⁺ metabolism in normal cells (not-supplemented) remains unclear.
2. Most experiments were carried out using cells/tissues that express higher level of Nr1 (liver or kidney cells). It is possible that in such cells/tissues, other NAD biosynthesis pathways are less active and therefore, utilization of NAM/NA in these cells appeared less efficient. The authors should specify that NR is a more efficient precursor only in these specific cell/tissue types unless additional studies in broader cell/tissue types are provided.
3. Comparing fig. 4a and 4c, it is not clear why NMN treatment resulted in more intracellular NR (fig. 4c) than direct NR treatment (Fig. 4a). Is it because extracellular NMN is more stable than NR?
4. In Figure 4, O18 carboxamide oxygen labeling of NR and NMN marks the nicotinamide moiety of NR and NMN and of course NAM itself. In this regard, detected NAM could directly come from NR (by nucleosidases) and not necessarily from NAD⁺ consumption reactions. If the ribose moiety of NR and NMN is labeled, the fractions of NR converted to either NAM or NMN can be determined.
5. What is the expression level of Nr2 in nr1KO cells? Perhaps, Nr2 expression is induced in nr1KO cells to compensate the loss of Nr1.
6. In Figure 6e, a fraction of extracellular (circulating) NMN and NR is converted to NAM, which suggested that NAMPT may play a role in NR/NMN supplementation induced increase in NAD⁺ levels. One possible experiment would be to examine the effect of NAMPT knockdown on NR/NMN induced NAD⁺ increase.

Reviewer #3 (Expert in NAD metabolism; Remarks to the Author):

The study by Canto, Brenner and colleagues is an important contribution to the field of NAD metabolism in mammals, which in recent years has emerged as an area of increasing interest due to its role in regulating energy metabolism, DNA repair, circadian rhythms, and aging. Some labs are working on NAD precursors as agents to prevent or treat numerous diseases in humans but how they are metabolized in mammals is poorly understood. The main question being addressed here is how nicotinamide riboside (NR) and nicotinamide mononucleotide (NMN) are metabolized. There are many discoveries of interest and novelty, the most impactful of which is the question of whether NMN is converted to NR outside the cell or within. It is known that NR is converted to NMN by NRK1/2. If NMN is directly transported into the cell, its conversion to NAD⁺ (a single step via the NMNATs) would not depend upon NRK1 activity. The data shows that NMN conversion to NAD⁺ in cells and mice does partly depend on NRK1. Overall the manuscript is well written but at times is over reaching, as stated below.

Main points:

1. The authors need to rule out that an NMN transporter is not downregulated by knockdown of NRK1 or an upregulation of CD38 and its homolog? This may also explain the data.
2. The strongest data is in vitro using labeled precursors to NAD. What is needed to prove

transport of NMN in the liver is not occurring to any great extent, they need to do animal 18O labeling experiment in vivo. It would also be important to test other tissues such as intestine, fat and muscle given that (i) tissues could be very different from one another and (ii) the work will contradict upcoming reports from the Imai lab that there is active NMN transport.

3. P6. Given this is going to be a contentious area, the study needs extra level of certainty. The authors state: "Thus, by transient overexpression, these data establish that increasing NRK activity is sufficient to boost NR-driven NAD⁺ synthesis, suggesting a rate-limiting role for NRK1 and/or NRK2 in NR utilization" is an example of a borderline proof at this stage of the paper. I can think of one example where they could be misled: the cells might be reducing CD38 levels or its homolog, as stated above.

4. In figure 6 h the authors show NR-injected NRK1 KO mice have a significant increase in circulating NR levels compared to vehicle treatment." Maybe I missed it, but did the authors test if NR and NMN plasma levels remain higher than WT if NRK is absent? That would be a prediction of the model.

5. Why is the effect only 60% when NRK is knocked out? Is the salvage pathway upregulated or CD38 downregulated in vivo?

6. The authors write: "However, NMN has proven challenging to detect in blood. Whereas NMN concentration has been reported to reach around 50 μ M in plasma (ref 13), our study and others (ref 12) have failed to detect comparable levels of circulating NMN. " This is tenuous ground. The Imai lab has stated (at least at conferences) that detection of NMN is column-dependent. Also, based on our lab's experience with NAD and its precursors, different HPLC-MS methods can give wildly different results depending on the column or solvent, it would seem premature to conclude that Imai is wrong. An independent method of verification would be prudent.

Minor:

7. The field will be interested to know that both NR and NMN injection also leads to significant increases of NAM in plasma, independent of genotype, given that NAM is a mammalian sirtuin inhibitor, the latter shown by Sinclair lab. That is worth discussing including implications for very high doses of either molecule.

8. p.6 To say that "The assay further confirmed that NRK1-D36A and NRK2-D35A mutants possess no kinase activity, as NR failed to increase NAD⁺ levels when these mutant forms were expressed." is a supposition based on an unproven fact at this stage of the manuscript. It should say "indicate" but also it would seem simple to test for a lack of in vitro activity.

9. "Further, hepatic NR increased in NRK1KO mice after administration of NR and modestly after NMN (Fig. 6g)." Why the difference?

10. What do the authors mean by the statement below? What is protein analysis? Please explain. "NR treatment led to significant increases in NAD content in HepG2, Hepa1.6 and AML12 cells (Fig. 3d, 3e and Supplementary Fig. 3b). Though our NRK1 antibody did not detect human NRK1, protein analyses in Hepa1.6 and AML12 confirmed a correlation between NRK1 levels and the ability of NR to enhance NAD synthesis.

11. As mentioned above, can the dispute in the field be reconciled by tissue specificity of an NMN transporter? This study mainly looked at liver and liver cells. Perhaps tissues vary. This should be discussed.

12. The authors write: "In sharp contrast, NRK1 is dispensable for conversion of NAM and NA into NAD, providing evidence that the effects of NR supplementation on NAD synthesis are not merely due to its degradation into NAM." Who claimed NAM was the source?

13. The authors write "Additionally, our data from quantitative mass spectrometry clearly show that NR supplementation does not lead to NAM accumulation in the media at any point before NAD synthesis. " (this is in vitro, so maybe in vivo it is different)

14. p.13: according to the authors, "the presence of NR in the human bloodstream is also supported by the fact that several bacteria affecting humans, such as Haemophilus influenza and Streptococcus pneumoniae rely on exogenous NR to support NAD synthesis and life." Being in fluids doesn't mean it is in blood. Please rewrite.

Reviewer #4 (Expert in NAD metabolism; Remarks to the Author):

The paper from Ratajczak et al. focuses on the enzyme NRK1 and provides evidence of the involvement of its catalytic activity in the conversion of NR to NAD. By overexpressing the gene in different cell lines (murine fibroblasts, murine and human hepatocytes) and by analyzing primary hepatocytes isolated from NRK1 KO mice, Authors show that in all tested cells NRK1 is rate-limiting and essential for NR and NMN utilization. Authors also show that in HepG2 cells extracellular NMN is dephosphorylated to NR prior to entering cell and be converted into NAD. Finally, they performed in vivo experiments on KO mice.

Although technically sound, this work presents results that mostly confirm previous data. Indeed, evidence on the essentiality of the enzyme in the conversion of NR to NAD in cultured cells already exist in the literature (Sociali et al, Oncotarget 2015), and, as also pointed out by the Authors, several studies have already indicated that in cultured cells extracellular NMN is transformed into NR prior to its utilization as NAD precursor. The only original result of this work is the direct evidence that NRK1 is the rate-limiting enzyme in the NAD biosynthetic route from NR.

Contrary to the data obtained in culture cells, the in vivo studies do not support two major conclusions of the work, i.e. the essentiality of NRK1 in the conversion of NR to NAD, and the requirement of conversion of extracellular NMN to NR in order to enter the cell and drive NAD synthesis. In fact, the in vivo experiments indicate that, at least in liver and kidney, NAD synthesis from NR can also occur through a NRK1-independent route, as also stated by the Authors. In addition, injected NMN is preferentially converted to NAM, rather than to NR in plasma (Fig. 6f, 6h), and administration of NMN does not increase at all hepatic NR in KO mice (Figure 6g), suggesting that injected NMN is preferentially converted to Nam rather than to NR. In this view, NAD synthesis from NMN and NR does not rely exclusively on NRK as Authors state both the title and the abstract.

Letter to the Reviewers:

We would like to thank the reviewers for taking the time to evaluate our manuscript and for their insightful comments. A point by point answer to their queries can be found below.

Reviewer #1:

1. *The method of cell harvesting could potentially affect cellular metabolism. It has been previously shown that trypsin affects central carbon metabolites (Badur, MG, Biotechnol. J.2015, Scoazec, M, Methods in Enzymology, 2014).*

The reviewer raises a very valid point. We have now evaluated the basal and NR-induced NAD⁺ levels in three different cell lines (HepG2, 3T3, F3T3-NRK1) using various cell collection methods: trypsinization, directly scraping the cells in PBS or treatment with 5 mM EDTA. The obtained results indicate that, in our hands, trypsin treatment leads to the most efficient NAD⁺ extraction, while extraction from scraped cells is the poorest (Figure 1 for Reviewers). Nevertheless, the relative increase in NAD⁺ level upon NR treatment remains unchanged regardless of the method used. Thus, these results illustrate that the use of trypsin is adequate for the purposes of our manuscript and that we would reach similar conclusions using other harvesting techniques.

2. *For Qqq analysis, was there an internal standard present in the samples to account for analytical variability?*

In figures 4 and 6 in which concentrations of analytes and the tracer/tracee ratios were reported, yes we did use internal standards. In figure 8, we performed relative quantification. However, we did account for analytical variability in two ways. First, we evaluated the repeatability of sample extraction procedure by a triplicate analysis of a pooled sample, obtaining relative standard deviations (%RSD, $n=3$) of 1%, 3% and 8% for NAM, NR and NMN respectively. Second, using a quality control (QC) made of a pool of all samples and alternatively inserted along the run sequence, we calculated the analytical/technical error/deviation so we can better estimate if biological differences are greater than analytical ones. As the referee can see in the table below, the measurements ($n=9$) were very robust, confirming that no significant degradation of NAD⁺ precursors and/or instrumental drift occurred during the sequence analysis.

	Liver				Plasma			
	Nam	NA	NR	NMN	Nam	NA	NR	NMN
Mean	4929749	2349	5747	628	218766	789	186	0
SD	50208	166	345	15	2399	69	25	0
CV	1.02%	7.06%	6.00%	2.46%	1.10%	8.68%	13.47%	N/A

Table 1. Standard deviation and coefficient of variation for LC-QqQ MS analysis of mouse liver and plasma samples.

3. *It would be useful to see the MID plots for the isotopologues formed from [18O]-NR and [18O]-NMN.*

We provide MID spectra for NR and NMN labelled compounds in Supplementary Fig. 6 as an evidence of identity, incorporation and fragmentation. We also provide the full spectra and the MS instrument/experiment details.

4. *It would also be good to have a schematic of the metabolic pathways.*

This is an excellent suggestion. We included the schematic with possible metabolic pathways in Figure 2c of the manuscript.

Reviewer #2:

1. *The title sentence is overstated. Nr1 is important but not essential for NR or NMN metabolism. As shown in Fig. 6a-6d, nr1KO liver and kidney cells were still able to convert a significant fraction of NR and NMN to NAD⁺. In addition, neither Nr1 over-expression nor nr1KO affects the basal level of NAD⁺ (Fig. 2d, 3d, 6a-d). The effects of Nr1 (either KO or overexpression) on NR or NMN utilization were only seen when cells (or mice) were treated with NR or NMN. Therefore, these results mainly justified the role of Nr1 in the utilization of supplemented NR/NMN. The role of Nr1 in NAD⁺ metabolism in normal cells (not-supplemented) remains unclear.*

As the referee notes, NRK1 knock-out or overexpression did not affect the basal NAD⁺ levels in studied models. However, we limited our *in vivo* experiments to young 5-6-weeks old mice and we cannot exclude that NRK1 is required for basal NAD⁺ levels maintenance in older mice or in response to stress conditions. Nevertheless, we understand the point of the reviewer, thus we

changed the title to “NRK1 controls nicotinamide riboside and nicotinamide mononucleotide metabolism in mammalian cells” in order to prevent any misinterpretation. It is known that damaged nerves and muscle cells induce NRK2, suggesting that there are endogenous conditions that make NR available. However, this was beyond the scope of this study. We set out to determine whether supplemented NR and NMN depend on NRK1 and have provided definitive and quantitative answers to this question. Though both compounds can be sources of NAM, extracellular NMN and extracellular NR form NAD metabolites largely through NRK1. Several additional experiments including quantitative flux analysis in order to reach this conclusion.

2. Most experiments were carried out using cells/tissues that express higher level of Nrk1 (liver or kidney cells). It is possible that in such cells/tissues, other NAD biosynthesis pathways are less active and therefore, utilization of NAM/NA in these cells appeared less efficient. The authors should specify that NR is a more efficient precursor only in these specific cell/tissue types unless additional studies in broader cell/tissue types are provided.

We agree. No such claim is made.

3. Comparing fig. 4a and 4c, it is not clear why NMN treatment resulted in more intracellular NR (fig. 4c) than direct NR treatment (Fig. 4a). Is it because extracellular NMN is more stable than NR?

This is a tracer:trace ratio and not a report of a concentration. Intracellular concentrations of NR are so low that it's hard to say much about them.

4. In Figure 4, O18 carboxamide oxygen labeling of NR and NMN marks the nicotinamide moiety of NR and NMN and of course NAM itself. In this regard, detected NAM could directly come from NR (by nucleosidases) and not necessarily from NAD+ consumption reactions. If the ribose moiety of NR and NMN is labeled, the fractions of NR converted to either NAM or NMN can be determined.

This is excellent point. We went the extra mile by custom synthesizing ^2H - ^{13}C -labeled NR and NMN in which both the nicotinamide (^{13}C) and ribose (^2H) moiety are labeled. We treated both wild type and NRK1KO-derived hepatocytes with these compounds. The results in Figure 6b and 6c clearly show that NR is able to enter the cells and lead to double-labelled NMN and NAD^+ only in WT hepatocytes (as seen by a significant increase in tracer/tracee ratio). Single labelled NAM was poorly detected extracellularly (Figure 2 for Reviewers), but equally in WT and NRK1 KO hepatocytes. Tests of NR stability show that, even after 6 hours in culture media, only 10-15% is degraded to NAM (Figure S4a). Strikingly, in this timecourse, double labeled NMN produced no intracellular signals as double labeled NMN or NAD.

5. What is the expression level of *Nrk2* in *nrk1KO* cells? Perhaps, *Nrk2* expression is induced in *nrk1KO* cells to compensate the loss of *Nrk1*.

This is also good point, as NRK2 compensation may potentially explain some of the results observed *in vivo*. However, in NRK1KO skeletal muscles we did not notice any differences in NRK2 expression (Supplementary Figure 3a). NRK2 in other tissues and in primary hepatocytes remained undetectable upon NRK1 deletion (Figure 5c).

In order to fully rule out the possible compensation of NRK2 in skeletal muscle, we evaluated the response to NR and NMN supplementation in NRK1/NRK2 double KO mice. The results obtained are comparable to these from NRK1 single KO, where NR and NMN administration still lead to NAD⁺ synthesis (Supplementary Figure 4b). This suggests that NRK2 does not compensate for NRK1 deficiency.

6. In Figure 6e, a fraction of extracellular (circulating) NMN and NR is converted to NAM, which suggested that NAMPT may play a role in NR/NMN supplementation induced increase in NAD⁺ levels. One possible experiment would be to examine the effect of NAMPT knockdown on NR/NMN induced NAD⁺ increase.

Following the referee's advice, we have subjected different cells to FK866 treatment to inhibit NAMPT activity and evaluated its influence on NR-, NMN- and NAM-induced NAD⁺ increase. We could observe that although FK866 treatment results in various decrease in basal NAD⁺ level in different cells, inhibition of NAMPT does not change the response to NR nor NMN supplementation (Supplementary Figure 4d). This suggest that, in cultured cell lines, the effects of NR and NMN are NAMPT-independent. This, together with our new data showing that NR is

degraded to NAM in mouse plasma, clearly indicates that the remaining action of NR and NMN after IP administration in NRK1 KO mice is due to NR conversion into NAM.

Reviewer #3:

1. The authors need to rule out that an NMN transporter is not downregulated by knockdown of NRK1 or an upregulation of CD38 and its homolog? This may also explain the data.

We have evaluated CD38 and CD157 protein expression in NRK1KO models and did not observe any differences (Figure 5c for primary hepatocytes and Supplementary Figure 3 for NRK1KO tissues), suggesting that changes in CD38 or CD157 levels are not responsible for the results observed in models of NRK1 deficiency. Similarly, we now show that CD73 expression in NRK1KO was not changed as well as PARP1 protein levels and activity, which further proves that the effects observed are truly due to NRK1 deficiency and not by changes in other NAD⁺-consuming activities.

Figure 6b was performed with wild-type primary hepatocytes. There was no intracellular double-labeled NMN when cells were incubated with double labeled NMN for 6 hours.

2. The strongest data is in vitro using labeled precursors to NAD. What is needed to prove transport of NMN in the liver is not occurring to any great extent, they need to do animal 18O labeling experiment in vivo. It would also be important to test other tissues such as intestine, fat and muscle given that (i) tissues could be very different from one another and (ii) the work will contradict upcoming reports from the Imai lab that there is active NMN transport.

We do not feel obligated to contradict data we cannot see. However, Trammell and co-workers have a paper in press at Nature Communications in which double-labeled NR was provided to mice by oral gavage and resulted in double-labeled NAD, NADP and NAAD. We directly tested predictions of NMN transport by double-labeling NMN. As shown in Figure 6b, double labeled NMN does not produce any double labeled intracellular metabolites in wild-type hepatocytes.

We also blocked utilization of NMN with CMP. We treated WT hepatocytes with CMP, which competitively inhibits extracellular nucleases transforming NMN into NR (Nikiforov et al., 2011). The results clearly illustrate that NMN, although perfectly active in control samples, cannot increase NAD⁺ in cells treated with CMP (Figure 6a). As expected, CMP treatment did not change cells response to NR.

Finally, to account for different NAD⁺ metabolism between tissues, we now also present data on skeletal muscle and BAT NAD⁺ levels in response to NR/NMN/NAM treatment *in vivo* (Figure 7e-f). Experiments in cultured cells also demonstrated that myotubes predominantly rely on NAM salvage in order to sustain NAD⁺ synthesis, explaining the poor response to NR or NMN *in vivo*.

3. Given this is going to be a contentious area, the study needs extra level of certainty. The authors state: "Thus, by transient overexpression, these data establish that increasing NRK activity is sufficient to boost NR-driven NAD+ synthesis, suggesting a rate-limiting role for NRK1 and/or

NRK2 in NR utilization" is an example of a borderline proof at this stage of the paper. I can think of one example where they could be misled: the cells might be reducing CD38 levels or its homolog, as stated above.

We now provide data not only on CD38, but also CD157, CD73 and NAMPT levels in F3T3-NRK1 and our overexpression systems (Supplementary Figure 1d). The modulation of NRK1 levels did not affect the expression of any of these proteins, further strengthening the concept of a rate-limiting role of NRK1.

4. In figure 6h the authors show NR-injected NRK1 KO mice have a significant increase in circulating NR levels compared to vehicle treatment." Maybe I missed it, but did the authors test if NR and NMN plasma levels remain higher than WT if NRK is absent? That would be a prediction of the model.

NR levels in mouse plasma in vehicle-treated animals do not show significant differences between genotypes. However, in whole blood, NR becomes rapidly cell-associated thereby preventing detection of differences in plasma levels. We did observe hepatic NR accumulation in NRK1KO mice after NR injection.

5. Why is the effect only 60% when NRK is knocked out? Is the salvage pathway upregulated or CD38 downregulated in vivo?

The levels of CD38 and other NAD⁺-consuming enzymes remain unchanged in NRK1KO deficient cells and tissues. The partial effect of KO *in vivo* may be explained by partial conversion of IP-injected NR and NMN into NAM (Figure 8b).

6. The authors write: "However, NMN has proven challenging to detect in blood. Whereas NMN concentration has been reported to reach around 50 mM in plasma (ref 13), our study and others (ref 12) have failed to detect comparable levels of circulating NMN." This is tenuous ground. The Imai lab has stated (at least at conferences) that detection of NMN is column-dependent. Also, based on our lab's experience with NAD and its precursors, different HPLC-MS methods can give wildly different results depending on the column or solvent, it would seem premature to conclude that Imai is wrong. An independent method of verification would be prudent.

We agree with the reviewer that NMN detection is column dependent. For this reason we applied a methodology based in hydrophilic liquid interaction chromatography (HILIC) on a UHPLC system, which provided a good sample separation and a good peak resolution and shape, minimizing matrix interferences and ion suppression due to matrix effects. Our detection limits were very low and similar for all determined NAD precursors (30nM for NAD, 4nM for NR and 20nM for NMN). Yet, it was not our goal to argue the validity of previous reports. In our hands NMN in plasma was detectable only at very low levels, far below the ones reported through HPLC. We now added a few lines in the discussion to acknowledge the variability of NMN detection between different methods (Page 15).

7. The field will be interested to know that both NR and NMN injection also leads to significant increases of NAM in plasma, independent of genotype, given that NAM is a mammalian sirtuin

inhibitor, the latter shown by Sinclair lab. That is worth discussing including implications for very high doses of either molecule.

We thank reviewer for the suggestion. More discussions on these lines have been added to the manuscript (see Page 14).

8. To say that "The assay further confirmed that NRK1-D36A and NRK2-D35A mutants possess no kinase activity, as NR failed to increase NAD⁺ levels when these mutant forms were expressed." is a supposition based on an unproven fact at this stage of the manuscript. It should say "indicate" but also it would seem simple to test for a lack of in vitro activity.

We agree with the reviewer and changed the text accordingly. However, we would like to highlight to the referee that the inactivity of these mutants has been previously defined and demonstrated (Tempel et al., 2007)

9. "Further, hepatic NR increased in NRK1KO mice after administration of NR and modestly after NMN (Fig. 6g)." Why the difference?

NR is directly transported into hepatic cells and can participate in hepatic NR pool as such, whereas NMN needs additional step of first being extracellularly dephosphorylated to NR. Due to this, NMN-derived hepatic NR levels may be much smaller than those coming directly from NR at the time point used in this experiment (1 hour after injection).

10. What do the authors mean by the statement below? What is protein analysis? Please explain. "NR treatment led to significant increases in NAD content in HepG2, Hepa1.6 and AML12 cells (Fig. 3d, 3e and Supplementary Fig. 3b). Though our NRK1 antibody did not detect human NRK1, protein analyses in Hepa1.6 and AML12 confirmed a correlation between NRK1 levels and the ability of NR to enhance NAD synthesis.

We changed the sentence to clearly indicate that we mean western blot analysis.

11. As mentioned above, can the dispute in the field be reconciled by tissue specificity of an NMN transporter? This study mainly looked at liver and liver cells. Perhaps tissues vary. This should be discussed.

We agree with the reviewer and for this reason we also show data for kidney, brown adipose and skeletal muscles (Figure 7). Moreover, our overexpression experiments (Figure 1 and 2) are performed in murine fibroblast cell line, NIH/3T3. As the putative NMN transporter has not been published yet, we are not able to compare the tissue-specific responses to NMN transporter expression profile. We added relevant discussion on this point to the manuscript.

12. The authors write: "In sharp contrast, NRK1 is dispensable for conversion of NAM and NA into NAD, providing evidence that the effects of NR supplementation on NAD synthesis are not merely due to its degradation into NAM." Who claimed NAM was the source?

Our *in vivo* data from NRK1KO mice injected intraperitoneally with NR and NMN may raise questions on why there is still NAD⁺ synthesis from these precursors in KO mice. As shown in Figure 8b, one possible explanation is a significant conversion of these compounds into NAM in

plasma of these mice. Notably, the remaining action of NR and NMN is very similar to that of NAM. With the sentence quoted by the referee, we wanted to highlight that NR action is not only an effect related to its conversion to NAM, but it indeed is a direct NR response. The Nature Communications paper by Trammell and co-workers also shows that NR and Nam have distinct effects on the hepatic NAD metabolome.

13. The authors write "Additionally, our data from quantitative mass spectrometry clearly show that NR supplementation does not lead to NAM accumulation in the media at any point before NAD synthesis." (this is in vitro, so maybe in vivo it is different)

We agree with reviewer and changed the text to clarify that this observation is valid in *in vitro* models.

14. p.13: according to the authors, "the presence of NR in the human bloodstream is also supported by the fact that several bacteria affecting humans, such as Haemophilus influenza and Streptococcus pneumoniae rely on exogenous NR to support NAD synthesis and life." Being in fluids doesn't mean it is in blood. Please rewrite.

We agree with the reviewer and changed the text accordingly.

Reviewer #4:

1. Although technically sound, this work presents results that mostly confirm previous data. Indeed, evidence on the essentiality of the enzyme in the conversion of NR to NAD in cultured cells already exist in the literature (Sociali et al, Oncotarget 2015), and, as also pointed out by the Authors, several studies have already indicated that in cultured cells extracellular NMN is transformed into NR prior to its utilization as NAD precursor. The only original result of this work is the direct evidence that NRK1 is the rate-limiting enzyme in the NAD biosynthetic route from NR.

Perhaps this reviewer should take this point up with other reviewers who still consider utilization of NMN an unresolved question.

The novelty of our manuscript relies on the following aspects:

- We provide for the first time genetic evidence on the need for NRK1 in order to metabolize NR and NMN in mammalian cell models and animal models
- We explore of the impact of NRK1 deletion in NAD⁺ homeostasis and response to NR precursors in up to 4 different tissues.
- We demonstrate the rate-limiting role of NR for the metabolism of exogenous NR and NMN
- Direct tracking of NR metabolism using stable-labelled tracer compounds. Furthermore, we show how only in cells expressing NRK1, double labelled tracers can lead to double labelled NAD⁺ intracellularly, certifying the key role of NRK1 in NR metabolism.
- We provide qualitative and quantitative information on NRK1 distribution and the relevance of NR metabolism in different tissues, in relation with other NAD⁺ precursors.

- We demonstrate how exogenous NR and NMN is transformed into NAM in circulation, and the dynamics of this effect. This has important consequences for the future and interpretations derived from NR and NMN supplementation or administration.

Given the above evidence, we consider that it would be unfair to limit the novelty and originality of this manuscript to the rate-limiting character of NRK1 on NR and NMN metabolism.

2. Contrary to the data obtained in culture cells, the in vivo studies do not support two major conclusions of the work, i.e. the essentiality of NRK1 in the conversion of NR to NAD, and the requirement of conversion of extracellular NMN to NR in order to enter the cell and drive NAD synthesis.

Concerning the essentiality of NRK1, we have amended the title in order to avoid any misinterpretation. All other aspects are more specifically responded below.

In fact, the in vivo experiments indicate that, at least in liver and kidney, NAD synthesis from NR can also occur through a NRK1-independent route, as also stated by the Authors.

The *in vivo* scenario is always more complex than the work on cultured cell lines. We provide evidence that this route is derived from NR transformation into NAM in plasma, a process that takes place in a NRK1-independent manner. This cleavage is largely unspecific and therefore does not remain the main mechanism for NR therapeutic actions. Supporting this, NAM supplementation has failed to display the constellation of benefits displayed by NR administration. Importantly, NMN is stable in plasma. Therefore, NAM appearance upon NMN intraperitoneal injection likely derives from its conversion to NR by cellular ectonucleotidases.

In addition, injected NMN is preferentially converted to NAM, rather than to NR in plasma (Fig. 6f, 6h), and administration of NMN does not increase at all hepatic NR in KO mice (Figure 6g), suggesting that injected NMN is preferentially converted to Nam rather than to NR.

We now present data indicating that NMN is not directly degraded to NAM in mouse plasma. Rather, NAM increases might be secondary to the transformation of NMN into NR. We also feel it would be adventurous to claim that NMN acts as an NAD⁺ precursor preferentially through conversion to NAM. Illustrating this, the action of NMN on NAD⁺ levels is blunted by more than 80% in kidney or brown adipose tissue from NRK1KO mice. Similarly, it is true that NMN did not lead to an increase in hepatic NR content, but it must be said that it also did not increase NAM. Therefore, the fact that NMN increases NAD⁺ without apparent increases in NR or NAM simply illustrates the tight equilibrium of multiple NAD⁺ metabolites intracellularly. Remarkably, NMN administration was followed by a tendency for an increase in intrahepatic NR ($p = 0.08$) in the NRK1KO livers, suggesting that NMN is actively transformed to NR and incorporated as NR in this tissue.

In this view, NAD synthesis from NMN and NR does not rely exclusively on NRK as Authors state both the title and the abstract.

The title and the abstract have been amended in order to tone down that statement, as the referee suggests.

References:

- Nikiforov, A., Dolle, C., Niere, M., and Ziegler, M. (2011). Pathways and subcellular compartmentation of NAD biosynthesis in human cells: from entry of extracellular precursors to mitochondrial NAD generation. *J Biol Chem* 286, 21767-21778.
- Tempel, W., Rabeh, W.M., Bogan, K.L., Belenky, P., Wojcik, M., Seidle, H.F., Nedyalkova, L., Yang, T., Sauve, A.A., Park, H.W., *et al.* (2007). Nicotinamide riboside kinase structures reveal new pathways to NAD⁺. *PLoS biology* 5, e263.

REVIEWERS' COMMENTS:

Reviewer #1 (Remarks to the Author):

The response to the comments are acceptable and I believe the paper is publishable as is.

Reviewer #2 (Remarks to the Author):

The authors have addressed most major concerns.

The reviewer is satisfied with the changes and additional results shown in this revised manuscript.

Reviewer #3 (Remarks to the Author):

The revision is markedly improved. The authors have performed extensive additional experiments that address the potential tissue-specificity of the transporter and ruled out effects on their metabolic enzymes CD157, CD73 and NAMPT. The text changes have also been made satisfactorily.

Reviewer #4 (Remarks to the Author):

Authors have addresses both concerns raised by this Reviewer on the in vivo experiments. They have amended both the title and the abstract to tone down the statements on the essentiality of NRK1.

They have performed additional experiments to support that in vivo most of extracellular NMN needs to be converted to NR in order to enter the cell and drive NAD synthesis.

In particular, Authors have demonstrated that, differently from NR, NMN is not directly degraded into Nam in plasma thus hypothesizing that injected NMN might be converted to NR by ectonucleotidases, and the formed NR is then degraded to Nam. Such an hypothesis is further supported by the clear evidence of a significant NMN-driven NAD synthesis through NRK1 in liver, kidney and BAT.

In general, this Reviewer agrees that the additional results in the revised paper together with the discussion throughout provide support to the role of NRK1 in the utilization of NMN also in vivo.

As for the novelty, this Reviewer wishes to reaffirm that evidence on the role of NRK in the utilization of exogenous NR and NMN for NAD biosynthesis in mammalian cultured cells has been already published by others. In this work, the Authors have provided additional and more direct evidence that strengthen the conclusion that extracellular NMN utilization to NAD cannot bypass NRK1 activity. Since, as also pointed by the Authors, conflicting data on NMN utilization are reported in the literature, the results presented in this work might significantly contribute to the field.

Letter to the Reviewers:

We would like to thank the reviewers for their enthusiastic, positive and constructive comments all along the reviewing process. We strongly believe that the referee's feedback has been instrumental to improve the quality of our studies, results and conclusions.